# Ecophysiology and interactions of a taurine-respiring bacterium in the mouse gut

Huimin Ye[1,2], Sabrina Borusak [3,14], Claudia Eberl[4,14], Julia Krasenbrink[1,2,14], Anna S. Weiss [4,14], Song-Can Chen[1], Buck T. Hanson[1,5,6], Bela Hausmann [7,8], Craig W. Herbold [1,13], Manuel Pristner[9], Benjamin Zwirzitz[1,5,6,12], Benedikt Warth [9,10], Petra Pjevac [1,7], David Schleheck[3], Bärbel Stecher[4,11] & Alexander Loy [1,7] ✉

Taurine-respiring gut bacteria produce H$_2$S with ambivalent impact on host health. We report the isolation and ecophysiological characterization of a taurine-respiring mouse gut bacterium. *Taurinivorans muris* strain LT0009 represents a new widespread species that differs from the human gut sulfidogen *Bilophila wadsworthia* in its sulfur metabolism pathways and host distribution. *T. muris* specializes in taurine respiration in vivo, seemingly unaffected by mouse diet and genotype, but is dependent on other bacteria for release of taurine from bile acids. Colonization of *T. muris* in gnotobiotic mice increased deconjugation of taurine-conjugated bile acids and transcriptional activity of a sulfur metabolism gene-encoding prophage in other commensals, and slightly decreased the abundance of *Salmonella enterica*, which showed reduced expression of galactonate catabolism genes. Re-analysis of metagenome data from a previous study further suggested that *T. muris* can contribute to protection against pathogens by the commensal mouse gut microbiota. Together, we show the realized physiological niche of a key murine gut sulfidogen and its interactions with selected gut microbiota members.

Hydrogen sulfide (H$_2$S) is an intestinal metabolite with pleiotropic effects, particularly on the gut mucosa[1,2]. H$_2$S can have a detrimental impact on the intestinal epithelium by chemically disrupting the mucus barrier[3], causing DNA damage[4], and impairing energy generation in colonocytes through inhibiting cytochrome *c* oxidase and beta-oxidation of short-chain fatty acids[5,6]. In contrast, low micromolar

concentrations of H$_2$S are anti-inflammatory and contribute to mucosal homeostasis and repair[7,8]. Furthermore, H$_2$S acts as a gaseous transmitter, a mitochondrial energy source, and an antioxidant in cellular redox processes. Thus, its impact on mammalian physiology and health reaches beyond the gastrointestinal tract[9]. For example, colonic luminal H$_2$S can promote somatic pain in mice[10] and contribute

[1]Division of Microbial Ecology, Centre for Microbiology and Environmental Systems Science, University of Vienna, Vienna, Austria. [2]Doctoral School in Microbiology and Environmental Science, Centre for Microbiology and Environmental Systems Science, University of Vienna, Vienna, Austria. [3]Department of Biology and Konstanz Research School Chemical Biology, University of Konstanz, Konstanz, Germany. [4]Max-von-Pettenkofer Institute, Ludwig Maximilian University Munich, Munich, Germany. [5]Austrian Competence Centre for Feed and Food Quality, Safety and Innovation FFoQSI GmbH, Tulln, Austria. [6]Institute of Food Safety, Food Technology and Veterinary Public Health, University of Veterinary Medicine, Vienna, Austria. [7]Joint Microbiome Facility of the Medical University of Vienna and the University of Vienna, Vienna, Austria. [8]Department of Laboratory Medicine, Medical University of Vienna, Vienna, Austria. [9]Department of Food Chemistry and Toxicology, University of Vienna, Vienna, Austria. [10]Exposome Austria, Research Infrastructure and National EIRENE Hub, Vienna, Austria. [11]German Center for Infection Research (DZIF), partner site Ludwig Maximilian University Munich, Munich, Germany. [12]Institute of Food Science, University of Natural Resources and Life Sciences, Vienna, Austria. [13]Present address: Te Kura Pūtaiao Koiora, School of Biological Sciences, Te Whare Wānanga o Waitaha, University of Canterbury, Christchurch, New Zealand. [14]These authors contributed equally: Sabrina Borusak, Claudia Eberl, Julia Krasenbrink, Anna S. Weiss. ✉e-mail: alexander.loy@univie.ac.at

to regulating blood pressure[11,12]. The multiple (patho)physiological functions of $H_2S$ in various organs and tissues depend on its concentration and the host's health status, but possibly also on the source of $H_2S$. Mammalian cells can produce $H_2S$ from cysteine via several known pathways[13]. In contrast to these endogenous sources, $H_2S$-releasing drugs and $H_2S$-producing intestinal microorganisms are considered exogenous sources. Sulfidogenic bacteria, which either metabolize organic sulfur compounds (e.g., cysteine) or anaerobically respire organic (e.g., taurine) or inorganic (e.g., sulfate, sulfite, tetrathionate) sulfur compounds in the gut, have a higher $H_2S$-producing capacity and are thus potentially harmful to their hosts[1,2,14]. Indeed, the abundance and activity of sulfidogenic gut bacteria were associated with intestinal diseases such as inflammatory bowel disease and colon cancer in various studies[15–17] and many gut pathogens, such as *Salmonella enterica* and *Clostridioides difficile*, are also sulfidogenic[18,19]. Excess bacterial $H_2S$ production combined with a reduced capacity of the inflamed mucosa to metabolize $H_2S$ is one of many mechanisms by which the gut microbiome can contribute to disease[20]. Yet, the manifold endogenous and microbial factors and processes that regulate intestinal $H_2S$ homeostasis are insufficiently understood.

A major and constantly available substrate of sulfidogenic bacteria in the gut is the organosulfonate taurine (2-aminoethanesulfonate). Taurine derives directly from the host's diet, in highest amounts from meat and seafood and to a lower extent from algae and plants[21], but is also liberated by microbial bile salt hydrolases (BSHs) from endogenously produced taurine-conjugated bile acids[22]. *Bilophila wadsworthia* is the most prominent taurine-utilizing bacterium in the human gut. Diets that contain high quantities of meat, dairy products, or fats can be associated with the outgrowth of *B. wadsworthia* in the gut[23,24]. Consumption of high-fat food triggers taurine-conjugated bile acid production and increases the taurine:glycine ratio in the bile acid pool[24]. In mouse models, higher abundances of *B. wadsworthia* can promote colitis and systemic inflammation[24,25] and aggravate metabolic dysfunctions[26].

*B. wadsworthia* metabolizes taurine via the two intermediates sulfoacetaldehyde and isethionate (2-hydroxyethanesulfonate) to sulfite[27], and the sulfite is utilized as electron acceptor for energy conservation and reduced to $H_2S$ via the DsrAB-DsrC dissimilatory sulfite reductase system[28]. The highly oxygen-sensitive isethionate sulfite-lyase system IslAB catalyzes the abstraction of sulfite (desulfonation) of isethionate[27,29]. Alternative taurine degradation pathways in other bacteria involve direct desulfonation of sulfoacetaldehyde by the oxygen-insensitive, thiamine-diphosphate-dependent sulfoacetaldehyde acetyltransferase Xsc[30–32]. Xsc is employed (i) in taurine utilization pathways of a wide range of aerobic bacteria that use taurine as carbon and energy source[31,33], and (ii) for anaerobic taurine fermentation by *Desulfonispora thiosulfatigenes*[32].

Here, we isolated a taurine-respiring and $H_2S$-producing bacterium from the murine intestinal tract and elucidated its fundamental and in vivo realized nutrient niche. We show that strain LT0009 represents a new *Desulfovibrionaceae* genus, for which we propose the name *Taurinivorans muris* gen. nov., sp. nov., and differs from its human counterpart *B. wadsworthia* by using the Xsc pathway for taurine degradation and its distribution across different animal hosts. We further provide initial insights into the protective role of *T. muris* against pathogen colonization.

## Results and discussion

### A taurine-respiring bacterium isolated from the murine gut represents a new genus of the family *Desulfovibrionaceae*

Strain LT0009 was enriched from mouse cecum and colon using an anoxic, non-reducing, modified *Desulfovibrio* medium with L-lactate and pyruvate as electron donors (and carbon source) and taurine as the sulfite donor for sulfite respiration. Its isolation was achieved by several transfers in liquid medium, purification by streaking on

ferric-iron supplemented agar plates, indicating sulfide production by black FeS formation and picking of black colonies, and by additional purification using dilution-to-extinction in liquid medium. We sequenced and reconstructed the complete LT0009 genome, which has a size of 2.2 Mbp, a G + C content of 43.6%, and is free of contamination as assessed by CheckM[34]. The genome comprises 2,059 protein-coding genes (Supplementary Data 1), 56 tRNA genes, 4 rRNA operons (with 5 S, 16 S, and 23 S rRNA genes), 4 pseudogenes, and 6 miscellaneous RNA genes.

LT0009 formed a monophyletic, genus-level (>94.5% similarity) lineage with other 16 S rRNA gene sequences from the gut of mice and other hosts. It has <92% 16 S rRNA gene sequence identity to the closest related strains Marseille-P3669 and *Mailhella massiliensis* Marseille-P3199[T] that were isolated from human feces (Fig. 1a). Phylogenomic treeing and an AAI of <60% to other described species strongly suggested that LT0009 represents the type strain of a novel genus in the family *Desulfovibrionaceae* of the phylum *Desulfobacterota*[35] for which we propose the name *Taurinivorans muris* (Fig. 1b, Supplementary Fig. 1, Supplementary Information). The previously described mouse gut MAGs UBA8003 and extra-SRR4116659.59 have >98% ANI and AAI to LT0009 and thus also belong to the species *T. muris*[36–38]. Furthermore, the mouse gut MAGs extra-SRR4116662.45 and single-China-D-Fe10-120416.2 showed 79% AAI/ANI to strain LT0009 and 84% AAI and 82% ANI to each other, which indicates that each of these two MAGs would likely represent a separate species in the genus *Taurinivorans*. Notably, the genome of *T. muris* LT0009 with only 2.2 Mbp in size is considerably smaller and thus potentially more streamlined than those of other free-living bacteria of the *Desulfovibrio-Bilophila-Lawsonia-Mailhella*-lineage (Fig. 1b). Only the obligate intracellular intestinal pathogen *Lawsonia intracellularis* has a smaller genome at 1.5-1.7 Mbp[39,40].

The Gram staining of *T. muris* LT0009 was negative. FISH imaging of the LT0009 pure culture with a genus-specific 16 S rRNA-targeted probe TAU1151 showed cells with a conspicuous spiral-shaped morphology and considerably varying lengths (1.7–28 μm) (Fig. 1c, Supplementary Data 2, Supplementary Fig. 2). SEM imaging further indicated that LT0009 cells have multiple polar flagella, suggesting that they are motile (Fig. 1c).

Complete utilization of taurine as electron acceptor in modified *Desulfovibrio* liquid medium with electron donors lactate and pyruvate in excess resulted in production of nearly quantitative amounts of $H_2S$ and excess acetate (Fig. 1d). Strain LT0009 in pure culture did not grow in the absence of 1,4-naphthoquinone and yeast extract, indicating an absolute requirement of menaquinone (vitamin K2) and other essential growth factors, respectively. Both growth rate and final growth yields were increased when taurine was provided at 20 and 40 mmol/l concentration in comparison to 10 mmol/l, while growth was inhibited at concentrations ≥60 mmol/l taurine (Supplementary Fig. 3a). Strain LT0009 grew with a lower growth rate and final growth yield when pyruvate was omitted as additional electron donor (Supplementary Fig. 3b). Strain LT0009 grew equally well at a pH range of 6–8.5 (Supplementary Fig. 3c) and temperatures between 27–32 °C, but with reduced final growth yield at 37 and 42 °C (Supplementary Fig. 3d). No colony formation was observed on agar plates under aerobic conditions, suggesting a strict anaerobic lifestyle of *T. muris* LT0009.

### Sulfur and energy metabolism of *T. muris* LT0009

Based on a genome-inferred metabolism prediction of strain LT0009 (Fig. 2a, Supplementary Data 3), we tested its growth with substrates that could serve as energy and sulfur sources in the gut. Fermentative growth with only pyruvate or only taurine, i.e., as both electron donor and acceptor, was not observed. In addition to pyruvate and lactate, LT0009 also used formate as electron donor for growth under taurine-respiring conditions, albeit with an extended lag phase and a lower

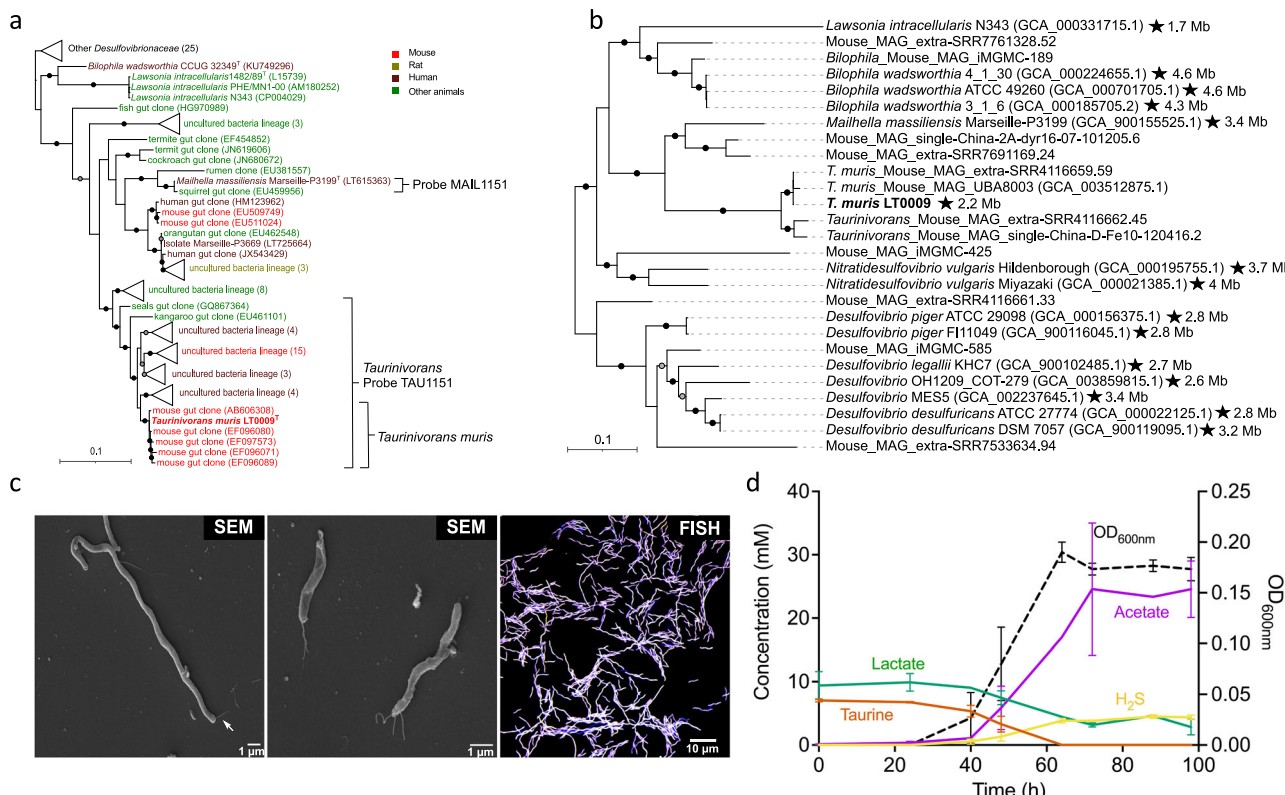

**Fig. 1 | Phylogeny and morphology of the mouse gut-derived taurine-respiring strain LT0009 that represents the new genus/species *Taurinivorans muris* in the family *Desulfovibrionaceae*. a** 16 S rRNA gene tree and FISH probe coverage. Maximum likelihood branch supports (1000 resamplings) equal to or greater than 95% and 80% are indicated by black and gray circles, respectively. The scale bar indicates 0.1 estimated substitutions per residue. Accession numbers are shown in parentheses. Strain LT0009 is shown in bold and type strains are marked with a superscript 'T'. Sequence sources are indicated with different colors (Supplementary Data 2). Sequences were assigned to *Taurinivorans* and *Taurinivorans muris* based on the genus-level similarity cutoff of 94.5% and species-level similarity cutoff of 98.7%[106], respectively. The perfect-match coverage of probes TAU1151 for *Taurinivorans* and MAIL1151 for *Mailhella* is indicated. **b** Phylogenomic tree. Ultrafast bootstrap support values equal to or greater than 95% and 80% for the maximum likelihood tree are indicated with black and gray circles, respectively. Accession numbers are shown in parentheses. Strain LT0009 is shown in bold.

Strains with complete genomes (genome size is indicated) are marked with a star. Genomes were assigned to *Taurinivorans* based on the genus-level AAI cutoff value of 63.4%[107]. The scale bar indicates 0.1 estimated substitutions per residue. **c** Representative morphology of LT0009 cells in pure culture (*n* = 3 independent cultures). SEM: Scanning electron microscopy images of cells of varying lengths. White arrows indicate the flagella. FISH: Cells hybridized with Cy3-labeled probe TAU1151 and Fluos-labeled probe EUB338mix and counterstained by DAPI. **d** Growth of strain LT0009 in modified *Desulfovibrio* medium confirmed complete utilization of taurine as electron acceptor concomitant with nearly stoichiometric production of sulfide. Electron donors L-lactate and pyruvate were provided in excess, and their utilization also contributed to acetate formation. Pyruvate in the medium and ammonia released from the deamination of taurine were not analyzed in this experiment. Lines represent averages of measures in triplicate cultures. Error bars represent one standard deviation (*n* = 3 biologically independent cultures). Source data are provided as a Source Data file.

growth yield. For the alternative electron acceptors tested, LT0009 used 3-sulfolactate and thiosulfate in combination with lactate and pyruvate, but did not grow with 2,3-dihydroxypropane-1-sulfonate (DHPS), isethionate, cysteate, and not with inorganic sulfate or sulfite (Fig. 2b).

The metabolic pathways used for growth by respiration with taurine, sulfolactate, or thiosulfate and with lactate/pyruvate as electron donors were further analyzed by both differential transcriptomics and differential proteomics, as transcription and translation can be uncoupled in bacteria[41,42]. These analyses and comparative sequence analyses of the key genes/proteins (Supplementary Fig. 4) strongly suggested that taurine is degraded via the Tpa-Ald-Xsc pathway and the produced sulfite respired via the DsrAB-DsrC pathway (Fig. 2c, d). Notably, Tpa, Ald, and Xsc were among the top thirteen most abundant proteins that were detected in strain LT0009 grown with all three electron acceptors (Supplementary Fig. 5). Yet, relative protein abundances of Tpa and Ald but not Xsc were significantly increased in cells grown with taurine compared to cells grown with thiosulfate (Fig. 2d). Pyruvate-dependent taurine transaminase Tpa catalyzes initial conversion of taurine to alanine and sulfoacetaldehyde[32]. Oxidative

deamination of alanine to pyruvate is catalyzed by alanine dehydrogenase Ald (Fig. 2a, c, d). Lack of the sulfoacetaldehyde reductase gene *sarD* and *islAB* and inability to grow with isethionate showed that LT0009 does not have the taurine degradation pathway of *B. wadsworthia*[27]. Instead, sulfoacetaldehyde is directly desulfonated to acetyl-phosphate and sulfite by thiamine-diphosphate-dependent sulfoacetaldehyde acetyltransferase Xsc[30-32]. The acetyl-phosphate is then converted to acetate and ATP by acetate kinase AckA. Strain LT0009 seems to lack candidate genes for the TauABC taurine transporter[43]. While homologs of *tauABC* are encoded in the genome, the individual genes do not form a gene cluster like in *Escherichia coli*[44] and were not expressed during growth on taurine (Supplementary Data 1). Instead, the LT0009 genome encodes three copies of gene sets for tripartite ATP-independent periplasmic (TRAP) transporter[45] that are co-encoded in the taurine degradation gene cluster and were expressed during growth with taurine, thus are most likely involved in taurine import, including DctPQM1 (with fused DctQM1) (TAU-VO_v1_1026 and 1027) and DctPQM3 (TAUVO_v1_1467-1469) (Fig. 2c, d).

(2 S)-3-sulfolactate is degraded by LT0009 via the SlcC-ComC-SuyAB pathway as shown by differential expression of these enzymes

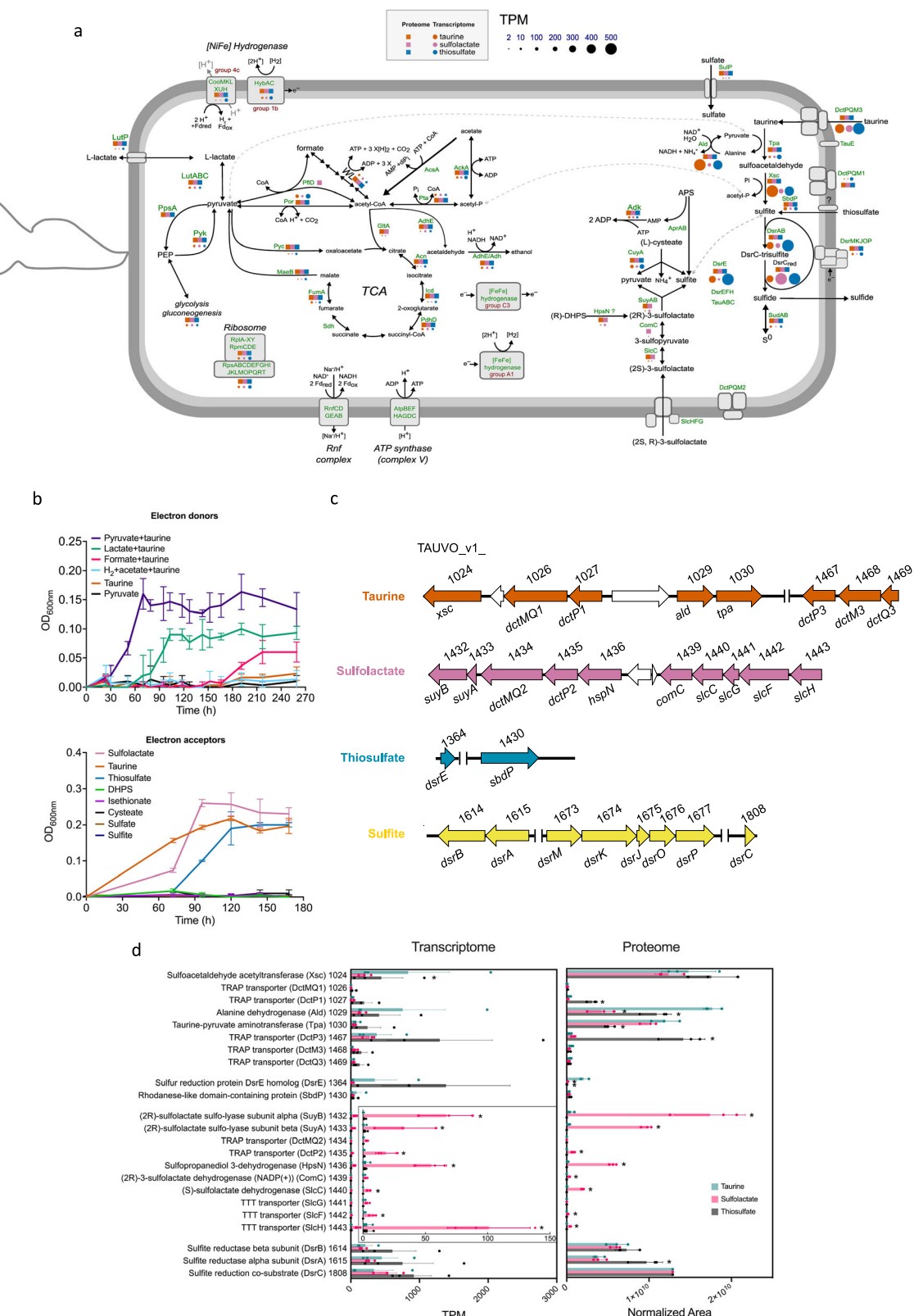

in cells grown with racemic sulfolactate (Fig. 2c, d, Supplementary Data 1). The dehydrogenases (S)-sulfolactate dehydrogenase SlcC and (R)-sulfolactate oxidoreductase ComC isomerize (2 S)-3-sulfolactate to (2 R)-3-sulfolactate via 3-sulfopyruvate. (2 R)-3-sulfolactate is desulfonated to pyruvate and sulfite by sulfo-lyase SuyAB, which was significantly increased in sulfolactate-grown cells compared to taurine or thiosulfate-grown cells (Fig. 2d). The neighboring gene clusters

*slcHFG-slcC-comC* and *hpsN-dctPQM2-suyAB* were both significantly upregulated in the transcriptome of sulfolactate-grown cells (Fig. 2d). The DctPQM2 (with fused DctQM2) (TAUVO_v1_1434 and 1435) TRAP transporter and the SlcGFH tripartite tricarboxylate transporters (TTT)[45,46] are putative sulfolactate importers. The gene cluster further includes a homolog to *hpsN*, encoding sulfopropanediol-3-dehydrogenase[47]. This enzyme converts (R)-DHPS to (R)-sulfolactate

**Fig. 2 | Sulfur-based energy metabolism of *Taurinivorans muris* LT0009. a** Cell cartoon of the central sulfur and energy metabolism of LT0009 as determined by genome, transcriptome, and proteome analyses. Genes/proteins detected in the transcriptome and proteome of LT0009 grown with taurine, sulfolactate, or thiosulfate as electron acceptor are shown by colored circles and squares, respectively. Circle size indicates gene transcription level normalized as TPM. Proteins of all transcribed genes were also detected in the proteome, with the exception of AprAB, TauA (TAU_v1_0027, TAU_v1_1344), TauB, TauC, TauE, DctMQ2, SlcG, DsrEFH, AscA, two [FeFe] hydrogenases (TAU_v1_1126, TAU_v1_1901), cytochrome *c* and Sdh. Protein complexes (e.g., Rnf, SlcFGH, DctPMQ2, Atp) are not shown with transcriptome and proteome data because at least one gene/protein of the complex units was not detected. The gene annotations are listed in Supplementary Data 7. **b** Anaerobic growth tests of strain LT0009 with various substrates (*n* = 3 biologically independent cultures). Electron donors: All substrates were added at 10 mmol/l concentration, except acetate (20 mmol/l), which was added as carbon source together with H$_2$. Electron acceptors: The different sulfur compounds were added at 10 mmol/l concentration together with pyruvate, lactate, and 1,4-naphthoquinone. OD$_{600}$: optical density at 600 nm. **c** Organization of sulfur metabolism genes in the LT0009 genome. Numbers show the RefSeq locus tag with the prefix TAUVO_v1. **d** Comparative transcriptome and proteome analysis of LT0009 grown with lactate and taurine, sulfolactate, or thiosulfate as electron acceptor. Numbers following protein names refer to RefSeq locus tag numbers (prefix TAUVO_v1). Protein expression was normalized to DsrC for each growth condition. Bars represent averages and error bars represent one standard deviation (*n* = 3 biologically independent cultures). Asterisk indicates significant (*p* < 0.05, exact *p* values are provided in Supplementary Data 1) differences in gene transcription/protein expression compared to growth with taurine (5% false discovery rate, DESeq2 Wald test). TCA, tricarboxylic acid cycle; WL, Wood-Ljungdahl pathway; PEP, phosphoenolpyruvate; DHPS, 2,3-dihydroxypropane-1-sulfonate; TPM, transcripts per million. Source data are provided as a Source Data file.

during aerobic catabolism of DHPS by diverse bacteria in soils[54] and the ocean[48]. However, LT0009 did not grow with racemic DHPS when tested (Fig. 2b). The *hpsN* gene was transcribed in LT0009 with taurine, sulfolactate, and thiosulfate treatments, but the HpsN protein was not detected (Supplementary Data 1). LT0009 did not grow with cysteate as an electron acceptor under the conditions we used, although it encodes a homolog of L-cysteate sulfo-lyase CuyA that desulfonates L-cysteate to pyruvate, ammonium, and sulfite[49,50] (Fig. 2b). Yet, CuyA can also act as D-cysteine desulfhydrase[49]. The *cuyA* gene was transcribed in the presence of taurine, sulfolactate, and thiosulfate. Furthermore, CuyA was significantly higher expressed in LT0009 with taurine (*P* < 0.001) compared with sulfolactate and thiosulfate (Supplementary Data 1), yet its physiological role in LT0009 remains unclear.

Strain LT0009 respired thiosulfate, such as *B. wadsworthia* strain RZATAU[51], but lacks genes for PhsABC thiosulfate reductase[52] and thiosulfate reductase from *Desulfovibrio* (EC 1.8.2.5)[53], which both (i) disproportionate thiosulfate to sulfide and sulfite and (ii) are present in the human *B. wadsworthia* strains ATCC 49260, 4_1_30, and 3_1_6. LT0009 has a gene for a homolog of rhodanese-like, sulfur-trafficking protein SbdP (TAUVO_v1_1430) (Supplementary Fig. 4g)[54]. Rhodaneses (EC 2.8.1.1.) can function as thiosulfate sulfur transferase and produce sulfite[55]. Homologs of SbdP are broadly distributed in members of the *Desulfovibrio-Bilophila-Mailhella-Taurinivorans* clade and other *Desulfovibrionaceae* (Supplementary Fig. 4g). The SbdP-homolog in LT0009 (TAUVO_v1_1430) could (i) provide sulfite for reduction and energy conservation by the Dsr sulfite reductase system and (ii) transfer the second sulfur atom from thiosulfate to an unknown acceptor protein/enzyme. A candidate sulfur-accepting protein is encoded by a *dsrE*-like gene in LT0009 (TAUVO_v1_1364) (Supplementary Fig. 4e). High expression of rhodanese-like sulfur transferases, a DsrE-like protein, and DsrAB sulfite reductase was reported for thiosulfate-respiring *Desulfurella amilsii*[56]. However, the functions of the SbdP-sulfur transferase and the DsrE-like protein in the thiosulfate metabolism of LT0009 remain unconfirmed. First, these proteins are not homologous to the highly expressed *D. amilsii* proteins. Second, comparative transcriptome and proteome analyses were inconclusive, as only the transcription of the *dsrE*-like gene was upregulated in LT0009 grown with thiosulfate (Fig. 2d, Supplementary Data 1). Additional genes homologous to known sulfur metabolism genes whose functions in LT0009 remain enigmatic include *sudAB*, which encode sulfide dehydrogenase for reduction of sulfur or polysulfide to H$_2$S[57], and *dsrEFH*, which are involved in sulfur atom transfer in sulfur oxidizers (Supplementary Fig. 4f)[58].

LT0009 encodes an incomplete pathway for dissimilatory sulfate reduction. While homologs of genes for the sulfate transporter SulP[59] and adenylyl-sulfate reductase AprAB were present, the absence of genes for sulfate adenylyltransferase Sat and the electron-transferring QmoAB complex (Supplementary Data 3) was consistent with the inability of LT0009 to respire sulfate. Although externally supplied sulfite did not support growth, the DsrAB-DsrC dissimilatory sulfite reductase system was highly expressed in cells grown with taurine, sulfolactate or thiosulfate (Fig. 2a, d, Supplementary Data 1). This suggests that intracellularly produced sulfite is respired to sulfide via the DsrAB-DsrC system, which includes the transfer of electrons from the oxidation of electron donors via the membrane quinone pool and the DsrMKJOP complex (Fig. 2a)[60].

Genome reconstruction of LT0009 suggested the potential to utilize lactate, pyruvate, and H$_2$ as electron donors (Supplementary Information). We experimentally confirmed that lactate, pyruvate, and formate are used as electron donors for taurine respiration (Fig. 2b). *T. muris* and *B. wadsworthia* use similar electron acceptors and donors, yet differ in the metabolic pathways to use them. Differential relative mRNA and protein abundances of *T. muris* LT0009 grown on taurine and sulfolactate suggest a differential regulation of the two organosulfonate metabolisms. However, strain LT0009 maintained a high cellular prevalence of its taurine metabolism enzymes Tpa, Ald, and Xsc independent of the used electron acceptor. Given that taurine is a largely host-derived and thus permanently available substrate in the animal gut, a predominantly constitutive expression and/or high stability of taurine-degrading proteins could sustain the fitness of *T. muris* in the gut ecosystem[61].

## Distinct distribution patterns of *Taurinivorans muris* and *Bilophila wadsworthia* suggest different host preferences

*B. wadsworthia* was repeatedly reported as a taurine-degrading member of the murine intestine based on molecular surveys[24,26,62]. We performed a meta-analysis to compare the presence and relative abundance of *B. wadsworthia*-related and *T. muris*-related sequences across thousands of 16 S rRNA gene amplicon datasets from the intestinal tract of diverse hosts. *T. muris*-related 16 S rRNA gene sequences were most often detected in the mouse gut, i.e., in 14.4% of all mouse amplicon datasets, but also present in the datasets from multiple other hosts (shrimp, pig, rat, chicken, fish, cow, humans, termites, and other insects) (Fig. 3a, Supplementary Data 4). In comparison, *B. wadsworthia*-related sequences are most widespread in the human gut, i.e., in 30.7% of human gut amplicon datasets, but are also prevalent in pig (15.8%), chicken (13.7%), and rat (9.8%) and occasionally detected in other hosts (Fig. 3a). We also identified *B. wadsworthia*-related sequences in 7.5% of mouse datasets. *T. muris*- and *B. wadsworthia*-related sequences co-occurred only in 28 mouse datasets, which could be due to differential adaptation to the murine and human gut of the two organisms and/or their competitive exclusion by competition for taurine. Furthermore, we found that 82% of the *B. wadsworthia*-positive samples are from mice that were 'humanized' by receiving human feces transplants or human strain consortia[63–66], which suggests a much lower prevalence of *B. wadsworthia* strains that are indigenous in mice. *T. muris*-related sequences represented >5% of

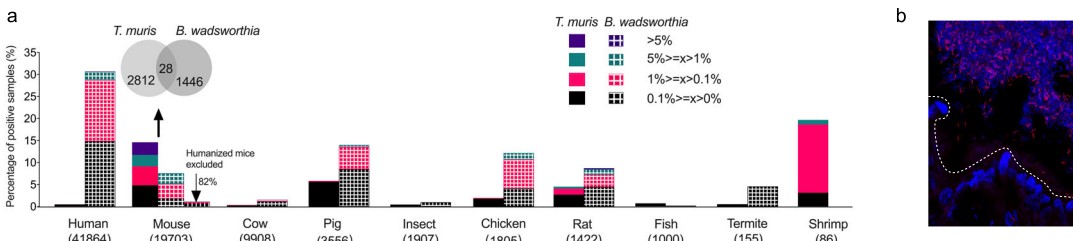

**Fig. 3 | *Taurinivorans muris* and *Bilophila wadsworthia* have distinct host distribution patterns. a** Occurrence and prevalence of *Taurinivorans muris*- and *Bilophila wadsworthia*-related sequences in 16 S rRNA gene amplicon datasets of human and animal guts. *T. muris*- and *B. wadsworthia*-like sequences at 97% similarity cutoff are expressed as percentages of positive samples in each host (the numbers of samples used for the analysis are shown in parenthesis) and different colors indicate percentages of samples positive for *T. muris* and *B. wadsworthia* at different relative abundance ranges. Hosts with less than 20 amplicon samples are not shown. *T. muris*- and *B. wadsworthia*-related sequences co-occur in only 28 mouse gut samples as shown by the Venn diagram. For comparison, the abundance of *B. wadsworthia*-positive mouse gut samples is also shown after removal of data from mice that were 'humanized' by receiving human feces transplants or human strain consortia. **b** Visualization of *Taurinivorans* in a colon tissue section of a mouse (*n* = 1) fed a polysaccharide- and fiber-deficient diet[108] by FISH. TAU1151-Cy3-labeled *Taurinivorans* cells appear in pink and the remaining bacterial cells and tissue in blue due to DAPI-staining. The dashed line indicates the border between epithelial cells and the gut lumen. Source data are provided as a Source Data file.

the total community in 2.8% of mouse gut datasets (Fig. 3a). Such very high relative 16 S rRNA gene abundances were more often observed in mice on high-fat diets[67,68], but sporadically also in mice on standard chow and other diets (Fig. 3b)[69].

Overall, *T. muris* is considerably more abundant and prevalent in the mouse gut microbiome than *B. wadsworthia*. Notably, a mouse native *B. wadsworthia* strain has not yet been isolated. Our phylogenomic analysis of all de-replicated, high-quality *Desulfovibrionaceae* MAGs from the integrated mouse gut metagenome catalog (iMGMC)[70] revealed two MAGs form a well-supported monophyletic group with *B. wadsworthia* strains (Fig. 1b). Mouse MAG iMGMC-189 has a minimum ANI of 82% and AAI of 79% to *B. wadsworthia*, which suggests that it represents the population genome of a new, murine *Bilophila* species. Mouse MAG extra-SRR7761328.52 is more distantly related and has a minimum ANI of 78% and AAI of 65% to *B. wadsworthia*. Both mouse MAGs encode the taurine degradation pathway (*tpa-sarD-islAB*) of *B. wadsworthia* (Supplementary Fig. 1), while the pathway for sulfolactate degradation (*slcC-comC-suyAB*) is absent in MAG extra-SRR7761328.52. In general, genes for utilization of diverse organosulfonates are widely and patchily distributed in the *Desulfovibrio-Bilophila-Mailhella-Taurinivorans* clade (Supplementary Fig. 1)[71]. Other mouse *Desulfovibrionaceae* that encode the capability for taurine respiration include (i) the *Desulfovibrio*-affiliated MAGs extra-SRR7533634.94 and iMGMC-585 with the *tpa-xsc* pathway and (ii) the *Mailhella*-related MAG extra-SRR7691169.24 with the *tpa-sarD-islAB* pathway (Supplementary Fig. 1).

## Taurine degradation is the main in vivo realized nutritional niche of *Taurinivorans muris*

We next performed metatranscriptome analysis and re-analyzed published metatranscriptome datasets of gut samples from different mouse models to reveal the metabolic pathways that are most expressed by *T. muris* in its murine host. In our gnotobiotic model, strain LT0009 or a mock control was added to germ-free mice stably colonized with the synthetic OMM[12] community, which does not encode *dsrAB-dsrC* for sulfite respiration (Fig. 4a). Strain-specific qPCR assays showed that ten OMM[12] strains and strain LT0009 colonized the mice (Fig. 4b). Consistent with previous studies, strains *A. muris* KB18 and *B. longum* subsp. *animalis* YL2 were not detected[72,73]. Colonization of LT0009 did not affect the abundance of other strains, which indicated that LT0009 occupied a free niche in the intestinal tract of this gnotobiotic mouse model. The taurine metabolism (*tpa, ald, xsc*) and sulfite reduction (*dsrAB, dsrC*) genes were in the top 5% expression rank of all LT0009 genes (Fig. 4c). In contrast, transcription of the putative thiosulfate transferase gene (*sbdP*) ranked at 17% and of

sulfolactate degradation genes (*suyAB, slcC, comC*) ranked from 62% to 88% of all LT0009 genes (Fig. 4c). LT0009-colonized mice showed a significant, 15-fold reduction in fecal taurine concentrations (Fig. 4d). *T. muris* LT0009 thus largely occupied the vacant taurine-nutrient niche in the intestinal tract of OMM[12] mice. Metatranscriptome analyses of intestinal samples from conventional laboratory mice on various diets (e.g., high-glucose; high-fat/low-carbohydrate; low-fat/high-carbohydrate) and with different genetic backgrounds (wild-type; plin2) also showed that taurine degradation and sulfite respiration were within the top 5% of expressed LT0009 genes (Supplementary Fig. 6).

Free taurine in the murine intestine largely derives from microbial deconjugation of host-derived taurine-conjugated bile acids[74,75]. LT0009 does not encode bile salt hydrolase (BSH) genes and thus likely depends on other gut microbiota members to liberate taurine from taurine-conjugated bile acids. BSH genes are encoded across diverse bacterial taxa in the human and mouse gut[74,76,77]. In agreement with previous studies of bile acid transformations in the OMM[12] model[78,79], we identified BSH genes in seven OMM[12] strains (Supplementary Data 5). BSH gene transcription increased in *Enterocloster clostridioformis* YL32, *Enterococcus faecalis* KB1, *Bacteroides caecimuris* I48, and *Muribaculum intestinale* YL27, and decreased in *Limosilactobacillus reuteri* I49, but not significantly (Supplementary Data 5 and 6). Bile acid deconjugation by some of these OMM[12] strains has been confirmed in vitro[79]. Specifically, *E. faecalis* KB1, *B. caecimuris* I48, *M. intestinale* YL27, and *Bifidobacterium animalis* YL2 tested positive for deconjugation of taurine-conjugated deoxycholic acid, while *E. clostridioformis* YL32 either tested negative or was inhibited by addition of bile acids. The down-regulation of the BSH gene in *L. reuteri* I49 is consistent with the in vitro deconjugation capacity of this strain for glycine-conjugated deoxycholic acid but not for taurine-conjugated deoxycholic acid[79] and the generally lower abundance of glycine-conjugated bile acids in rodents[75]. We thus hypothesized that taurine degradation by LT0009 could provide a positive feedback mechanism on the expression of BSHs for taurine-conjugated bile acid deconjugation in the OMM[12] model. In support of this hypothesis, the abundance of several taurine-conjugated bile acids was significantly reduced in feces of OMM[12] mice with LT0009 (Fig. 4d). Additional in vitro growth experiments and mono- and co-colonization experiments with germ-free mice confirmed that *T. muris* LT0009 can not grow on taurine-conjugated bile acids in pure culture and is strictly dependent on bile acid-deconjugating bacteria for successful colonization of the mouse gut (Supplementary Information, Supplementary Figs. 7 and 8).

Thiosulfate is presumably a constantly present electron acceptor for microbial respiration in the mammalian gut as it is generated by

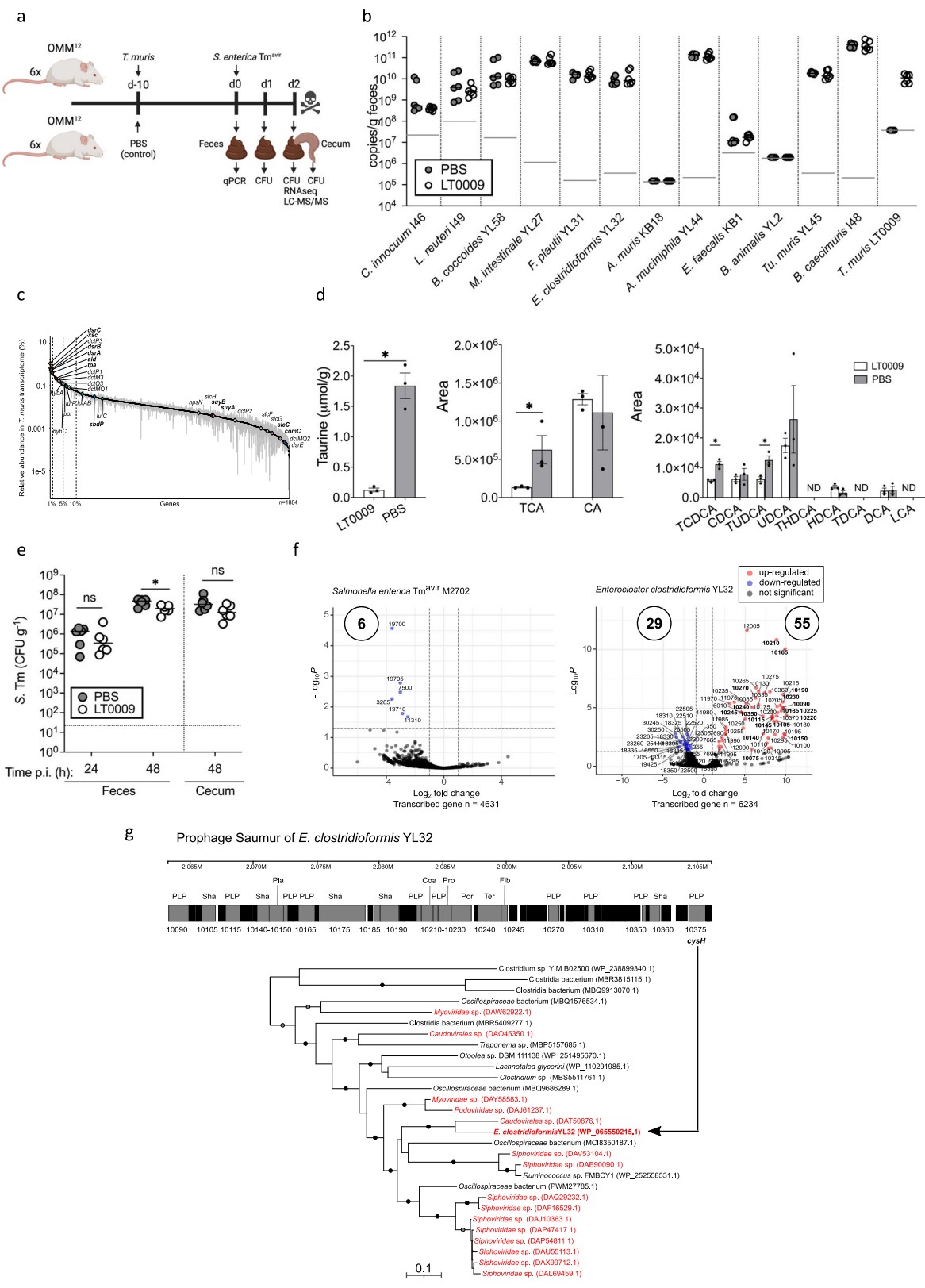

mitochondrial $H_2S$ oxidation in the gut epithelium[1]. Mitochondrial sulfide oxidation mainly takes place apically in the crypts of human colonic tissue at the interface to the gut microbiota[80]. The amount of thiosulfate supplied into the gut lumen depends on epithelial $H_2S$ metabolism[1,80]. The expression level of the putative SbdP thiosulfate transferase of *T. muris* ranked relatively high with 15–24% of all LT0009 genes across all gut samples from the various mouse models (Fig. 4c,

Supplementary Fig. 6). However, the function of this protein remains unconfirmed as its expression was not differentially upregulated in the thiosulfate-metabolizing *T. muris* pure culture (Fig. 2a, d, Supplementary Data 1). In vivo, taurine respiration, and potentially thiosulfate respiration, are likely fueled by pyruvate, $H_2$, and lactate as electron donors, as expression of genes for their oxidation ranked at 5–7% (*por*), 1.5–9.1% (*hybAC*), and 2.5–31% (*lutABC*, *lutP*) of all LT0009 genes across

**Fig. 4 | *Taurinivorans muris* mainly respires taurine released from taurine-conjugated bile acids by other bacteria and slightly enhances colonization resistance against *Salmonella enterica* in a gnotobiotic mouse model.**
**a** Schematic outline of the gnotobiotic mouse experiment. Created with Biorender.com. Mice stably colonized with the 12-strain Oligo-Mouse-Microbiota (OMM[12]) were inoculated with *T. muris* LT0009 ($n = 6$ mice) or sterile phosphate-buffered saline (PBS) as control ($n = 6$ mice) and, after 10 days, orally and rectally infected with *S. enterica* Tm[avir] M2702. Mice were sacrificed two days post infection (p.i.). Fecal samples were used for strain-specific 16 S rRNA gene-targeted quantitative PCR (qPCR). Fecal and cecal samples at 24 h and 48 h p.i. were used to analyze colony forming units (CFU) of *S. enterica* Tm. Fecal samples at 48 h p.i. were used for metatranscriptomics (RNAseq) and taurine and bile acids quantification (LC-MS/MS). **b** Absolute abundances (16 S rRNA gene copy numbers per gram feces) of each OMM[12] strain and strain LT0009 on day 10 in feces of mice with and without LT0009 (PBS). Small horizontal lines indicate median values. Gray horizontal lines indicate the detection limit of each strain-specific qPCR assay. **c** Ranked relative transcript abundance of LT0009 genes in OMM[12] mice fecal metatranscriptomes. Each point is the mean relative abundance of a gene and error bars correspond to the 95% confidence interval of the mean ($n = 3$ mice). The total number of transcribed LT0009 genes is shown ($n = 1884$). Genes for taurine (*tpa, xsc, ald*), sulfite (*dsrAB, dsrC*), sulfolactate (*suyAB, sclC, comC*) thiosulfate (*sbdP, dsrE*), pyruvate (*por*), lactate (*lutABC*), and hydrogen (*hybA, hybC*) metabolism are shown in different colors. Sulfur metabolism genes are further highlighted in bold font. Vertical dashed lines delineate the top 1%, 5%, and 10% expression rank of all 2059 protein-coding genes in the LT0009 genome. **d** Absolute concentration of taurine in feces of OMM[12] mice treated with LT0009 ($n = 3$ mice) or PBS ($n = 3$ mice). Semi-quantitative analyses of high abundant (TCA and CA) and low abundant bile acids. Colonization of LT0009 significantly reduced the concentrations of taurine ($p = 0.01$), TCA ($p = 0.04$), TCDCA ($p = 0.005$), and TUDCA ($p = 0.02$) in the feces of OMM[12] mice. THDCA, TDCA, and LCA were not detected. Mean values ± SD are plotted. *$p < 0.05$, Student's $t$ test, two-sided. CA, cholic acid; CDCA, chenodeoxycholic acid; UDCA, ursodeoxycholic acid; HDCA, hyodeoxycholic acid; DCA, deoxycholic acid; LCA, lithocholic acid; prefix T indicates taurine-conjugated bile acid species; ND, not detected. **e.** CFU of *S. enterica* Tm at 24 h and 48 h p.i. in the feces and at 48 h p.i. in the cecal content. Small horizontal lines indicate median values. The dotted horizontal line shows the CFU detection limit. The asterisk indicates significant differences ($p = 0.01$; student $t$ test, two-sided) between S. enterica Tmavir CFU in mice with LT0009 and the PBS-control mice at 48 h. ns, not significant. **f.** Volcano plots of differential gene transcription (5% false discovery rate, DESeq2 Wald test) of *S. enterica* Tm[avir] M2702 and E. clostridioformis YL32 in OMM[12] mice with ($n = 3$ mice) and without LT0009 ($n = 3$ mice). The $x$ axis shows log-fold-change in transcription and the $y$ axis shows the negative logarithm10-transformed adjusted $p$ values. Blue dots show significantly downregulated genes (adjusted $p$ value < 0.05, log2 fold change < −1) in mice with LT0009 and are labeled with locus tag numbers. Upregulated *E. clostridioformis* YL32 prophage genes in I5Q83_10075-10390 are highlighted in bold. **g.** Structure of the activated prophage gene cluster of *E. clostridioformis* YL32 and phylogeny of its encoded phosphoadenosine-phosphosulfate reductase (CysH). Virus- and bacteria-encoded sequences are shown in red and black, respectively. The maximum likelihood CysH tree is midpoint rooted. Ultrafast bootstrap support values equal to or >95% and 80% for the maximum likelihood tree are indicated with black and gray circles, respectively. The identity of the 61 genes in the prophage region (I5Q83_10075-10390) of *E. clostridioformis* YL32 as predicted by PHASTER[99]. Genes encoding hypothetical proteins are in black and annotated genes are in gray. Numbers indicate the locus tag. PLP phage-like protein, Sha tail shaft, Pla plate protein, Coa coat protein, Pro protease, Por portal protein, Ter terminase, Fib fiber protein. Source data are provided as a Source Data file.

all mouse gut metatranscriptomes, respectively (Fig. 4c, Supplementary Fig. 6).

Together with the highly specialized fundamental physiology of the *T. muris* pure culture (Fig. 2), these results strongly suggest that taurine is the predominant electron acceptor for energy conservation of *T. muris* in the murine intestinal tract.

### *Taurinivorans muris* LT0009 slightly increased colonization resistance against *S. enterica* and increased transcriptional activity of a sulfur metabolism gene-encoding prophage in a gnotobiotic mouse model

The human enteropathogen *S. enterica* Tm can invade and colonize the intestinal tract by utilizing various substrates for respiratory growth that are available at different infection stages[81]. The gnotobiotic OMM[12] mouse model provides intermediate colonization resistance against *S. enterica* Tm[72] and is widely used as a model system of modifiable strain composition for investigating the causal involvement of cultivated mouse microbiota members in diverse host diseases and phenotypes[82]. Yet, a bacterial isolate from the mouse gut with proven dissimilatory sulfidogenic capacity was unavailable until now[83]. *T. muris* has fundamental physiological features that could, on the one hand, contribute to colonization resistance against *S. enterica* Tm by direct competition for pyruvate[84], lactate[85], H₂[86], formate[87], and host-derived thiosulfate[87,88]. On the other hand, *T. muris* could also promote *S. enterica* Tm expansion during inflammation by fueling tetrathionate production through enhanced intestinal sulfur metabolism[18,89].

Here, we investigated the impact of *T. muris* LT0009 during the initial niche invasion of *S. enterica* Tm using an avirulent, non-colitogenic strain[90]. Compared to OMM[12] mice without LT0009, mice colonized with the OMM[12] and LT0009 had a slightly reduced load of *S. enterica* Tm at 48 p.i. that was significant in feces but not in cecum (Fig. 4e). Comparative metatranscriptome analysis did not provide evidence for H₂S-mediated inhibition of *S. enterica* Tm respiration, as shown for other enteropathogens[91], or any other mechanism of direct interaction. Six *S. enterica* Tm genes were differentially expressed, i.e., significantly downregulated, in the presence of strain LT0009 (Fig. 4f,

Supplementary Data 6). Three of the six genes are involved in transport and metabolism of galactonate (D-galactonate transporter DgoT, D-galactonate dehydratase DgoD, 2-dehydro-3-deoxy-6-phosphogalactonate aldolase DgoA), which is produced by some bacteria as an intermediate in D-galactose metabolism and is also present in mammalian tissue and body secretions[92]. D-galactonate catabolism capability was suggested as a distinguishing genetic feature of intestinal *Salmonella* strains compared to extraintestinal serovars, with serovars Typhi, Paratyphi A, Agona, and Infantis lacking genes for utilizing D-galactonate as a sole carbon source[93]. The putative D-galactonate transporter DgoT in *Salmonella enterica* serovar Choleraesuis was identified as a virulence determinant in pigs[94]. The OMM[12] strains and LT0009 do not encode the DgoTDAKR galactonate pathway. The significance of galactonate for *S. enterica* Tm gut colonization and competition remains to be elucidated.

Colonization of *T. muris* LT0009 in the gnotobiotic mice had variable impact on the differential gene expression pattern of the OMM[12] members (Fig. 4f, Supplementary Fig. 9, Supplementary Data 6). While gene expression was not significantly affected in *L. reuteri* I49, *E. clostridioformis* YL32 was most affected with 84 differentially expressed genes (55 upregulated and 29 downregulated) (Fig. 4f). Most of the significantly upregulated genes ($n = 41$) in *E. clostridioformis* YL32 are clustered in a large genomic region (I5Q83_10075-10390) that encoded various phage gene homologs and was identified as a prophage using PHASTER (Fig. 4g). This prophage, named YL32-pp-2.059, Saumur, is among a set of thirteen previously identified prophages of the OMM[12] consortium that represent novel viruses, were induced under various in vitro and/or in vivo conditions, and constitute the temporally stable viral community of OMM[12] mice[95]. We show that colonization of LT0009 in the OMM[12] mouse model selectively enhanced the transcriptional activity of the *E. clostridioformis* YL32 prophage Saumur, which carries a gene (I5Q83_10375) for phosphoadenosine-phosphosulfate reductase (CysH) that functions in the assimilatory sulfate reduction pathway of many bacteria (Fig. 4g). Various organosulfur auxiliary metabolic genes, particularly *cysH*, are widespread in environmental and

human-associated viromes, which suggests viruses augment sulfur metabolic processes in these environments, including the gut[96]. Addition of sulfide to a *Lactococcus lactis* strain culture was shown to increase production of viable particles of its phage P087[96], but it is unknown if H$_2$S can activate prophages. We speculated that *T. muris* LT0009 could impact intestinal sulfur homeostasis not only via its own sulfur metabolism but also by activation of the sulfur metabolism gene-expressing phage Saumur in *E. clostridioformis* YL32, possibly via H$_2$S. Growth tests with the pure culture of *E. clostridioformis* YL32 at two physiologically relevant sulfide concentrations demonstrated reduced growth at a high sulfide concentration (Supplementary Information, Supplementary Fig. 10). However, quantification of host and prophage genes did not show activation of the prophage Saumur under the tested condition. How prophage Saumur is activated and if its in vivo activity contributes to protection from *S. enterica* Tm thus remains subject of further study.

### *Taurinivorans muris* is the dominant sulfidogen in a wild-mouse-microbiota mouse model that provided H$_2$S-mediated protection against *Klebsiella pneumoniae*

Expansion of sulfidogenic bacteria and the *tpa-xsc-dsr* pathway in the metagenome fueled by host-derived taurine was previously shown to contribute to protection against the enteropathogen *Klebsiella pneumoniae* and *Citrobacter rodentium* in different mouse models; with sulfide-mediated inhibition of aerobic respiration by pathogens being proposed as a generic protective mechanism[91]. Here, we have re-analyzed the 16 S rRNA gene amplicon data and the reads of *dsrAB*, the sulfite reductase marker genes for sulfide production[97], from the metagenome data of this study. We identified *T. muris* as the dominant deltaproteobacterium (*Desulfobacterota*) and *dsrAB*-containing member of the microbiota of wild mice and the wild-mouse-microbiota (wildR) mouse model, but not in the other mouse models (Fig. 5). Given that taurine respiration via the sulfidogenic *tpa-xsc-dsr* pathway is the main energy niche of *T. muris* in the mouse gut (Fig. 4c, Supplementary Fig. 6), the enhanced resistance against *Klebsiella pneumoniae* in the wildR mouse model[91] was thus primarily due to the activity of *T. muris*. Sulfide-mediated protection against *Klebsiella pneumoniae* and *Citrobacter rodentium* in the other mouse models, i.e., taurine-supplemented and ΔyopM mice, was not provided by *T. muris* but by other sulfidogens (Fig. 5)[91].

### *Taurinivorans muris* LT0009 provides a core function in gnotobiotic mouse models

Dissimilatory sulfur metabolism with the production of H$_2$S is a core metabolic capability of the mammalian gut microbiota that is carried out by specialized bacteria[71,98]. As in humans and other animals, bacteria of the phylum *Desulfobacterota* (formerly *Deltaproteobacteria*), specifically of the *Desulfovibrio-Mailhella-Bilophila* lineage[35], are important sulfidogens in the mouse gut and appear to be more abundant in wild mice compared to untreated laboratory mice[99,100]. However, the diversity of sulfidogenic microorganisms and their metabolic pathways in the intestinal tract of non-human animals, including mice that represent important experimental models, remain insufficiently understood. A mouse gut-derived *Desulfobacterota* strain (*Desulfovibrio* strain MGBC000161) was recently isolated, but not physiologically characterized[99]. Here, we contribute the sulfidogenic strain LT0009, representing the newly proposed genus *Taurinivorans*, to the growing collection of publicly available bacterial strains from the mouse[83,99]. We further describe the fundamental metabolic properties and realized lifestyle of *Taurinivorans muris*, which relies on taurine as the primary electron acceptor for energy conservation in vivo and can contribute to the protective effect of the commensal mouse gut microbiota against *S. enterica* and *K. pneumoniae* (Fig. 6). In addition to direct inhibition of aerobic respiration of pathogens by H$_2$S as shown previously[91], indirect, not yet fully resolved resistance

mechanisms, which are dependent on the respective pathogen and composition of the resident microbiota, can be at play. Presence of *T. muris* selectively increased transcription of a phosphoadenosine-phosphosulfate reductase (CysH)-encoding prophage in another bacterium and could thereby modulate microbiome functions. *T. muris* strain LT0009 is currently the only murine *Desulfobacterota* isolate with a physiologically proven dissimilatory sulfur metabolism and thus significantly extends the experimental options to study the role of sulfidogenic bacteria in gnotobiotic mouse models.

## Methods

Supplementary Information provides further details on the methods described below. All animal experiments complied with ethical regulations and were approved by local authorities in Germany (Regierung von Oberbayern; ROB-55.2-2532.Vet_02-20-84) or by national Austrian authorities (Bundesministerium für Bildung, Wissenschaft und Forschung; BMWF-66.006/0032-WF/V/3b/2014).

### Isolation of strain LT0009 and growth experiments

Intestinal contents of wild-type C57BL/6 mice were used as inocula for the enrichment cultures. A modified *Desulfovibrio* medium was used to isolate strain LT0009 with taurine as electron acceptor and lactate and pyruvate as electron donors and for further growth experiments. Consumption of taurine and lactate and production of H$_2$S and SCFA were measured[71]. Strain LT0009 was tested for optimal taurine concentration and growth with/without pyruvate supplementation, different electron donors and acceptors, optimal pH and temperature, and growth on taurocholic acid.

### Microscopy

Gram staining of the LT0009 isolate was performed using a Gram staining kit according to the manufacturer's instruction (Sigma Aldrich, 77730-1KT-F), and its cellular morphology was imaged with a scanning electron microscope (JSM-IT300, JOEL). A probe was designed, tested, and applied for fluorescence in situ hybridization (FISH)-based microscopy of the genus *Taurinivorans* (Supplementary Data 2, Supplementary Fig. 2).

### Genome sequencing and comparative sequence analyses

The complete genome of strain LT0009 was determined by combined short- (Illumina) and long-read (Nanopore) sequencing. The automated annotation of the genome was manually curated for genes of interest, focusing on energy metabolism. Phylogenomic analyses comprised treeing with 43 concatenated marker sequences[34] and calculating average amino acid identities (AAI) and whole-genome average nucleotide identities (gANI). Additional phylogenetic trees were calculated with LT0009 using sequences of the 16 S rRNA gene and selected sulfur metabolism proteins or genes. Source information of 16 S rRNA gene reference sequences was manually compiled from the National Center for Biotechnology Information (NCBI) Sequence Read Archive (SRA) entries (Supplementary Data 7).

### Differential proteomics and transcriptomics

The total proteomes and transcriptomes of strain LT0009 grown with taurine, sulfolactate, or thiosulfate as electron acceptor and lactate/pyruvate as electron donor were determined and compared.

### Analyses of publicly available 16 S rRNA sequence data

The occurrence and prevalence of *Taurinivorans muris*- and *Bilophila wadsworthia*-related 16 S rRNA gene sequences were analyzed across 123,723 amplicon datasets from gut samples, including 81,501 with host information. Further information on mouse studies with at least 20 samples that were positive for *B. wadsworthia* was manually compiled from the NCBI SRA entries or the corresponding publications (Supplementary Data 8).

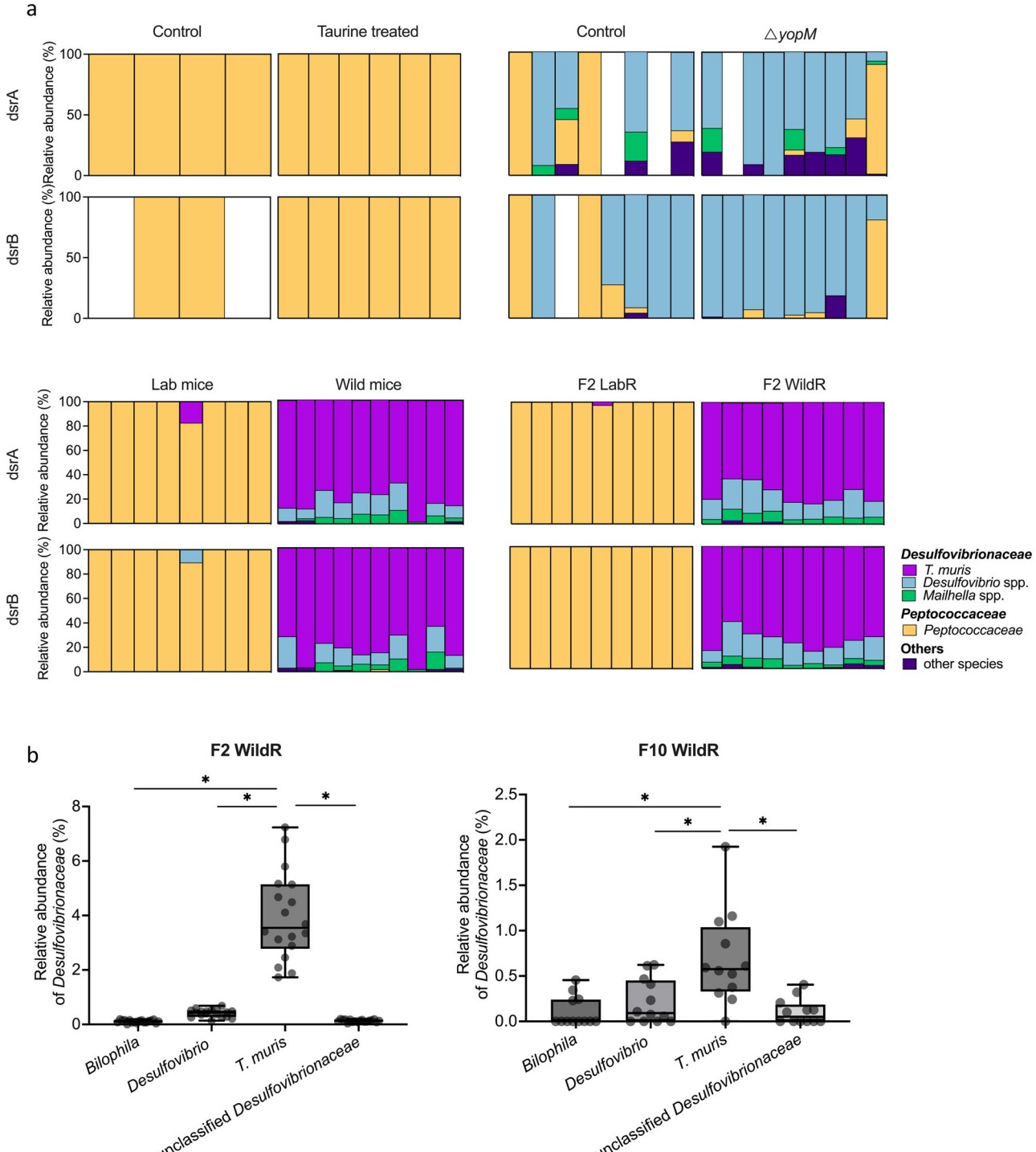

**Fig. 5 | *Taurinivorans muris* is the dominant sulfidogen in wild-mouse-microbiota-derived wildR mice that provided H₂S-mediated protection against *Klebsiella pneumoniae*.** Re-analysis of mouse gut metagenome and 16 S rRNA gene amplicon data from Stacy et al.[37]. **a** Identity and relative abundance of *dsrAB*-encoding taxa in different mouse models. Relative abundance was calculated with mapped *dsrA* and *dsrB* read counts. Samples with less than ten total mapped read counts were not displayed. Each column shows a sample from an individual mouse. Taurine-treated mice: specific-pathogen-free (SPF) mice that received taurine in drinking water and showed enhanced resistance to *Klebsiella pneumoniae* and *Citrobacter rodentium*; ΔyopM mice: SPF mice that were previously infected with the

attenuated strain ΔyopM of the food-borne pathogen *Yersinia pseudotuberculosis* and showed enhanced resistance to *K. pneumoniae*; SPF mice were used as control for taurine-treated and ΔyopM mice; lab mice: laboratory SPF mice; wild mice: wild-caught mice; F2 LabR mice: the second generation offspring of SPF mice whose germ-free founders received the microbiota of labR mice; F2 wildR mice: the second generation offspring of SPF mice whose germ-free founders received the microbiota of wild mice and showed enhanced resistance to *K. pneumoniae*.
**b** Relative 16 S rRNA gene abundance of *Desulfovibrionaceae* species in the 2nd (animal. Ordinary one-way ANOVA with Holm-Sidak's multiple comparisons test, *$p < 0.05$. Source data are provided as a Source Data file.

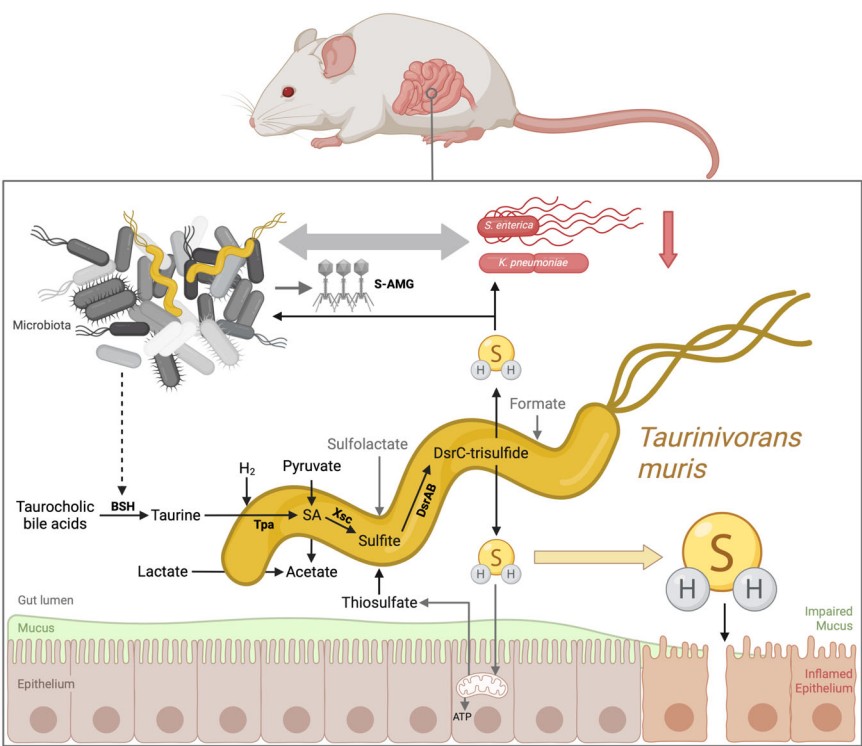

**Fig. 6 | Sulfur energy metabolism and proposed interaction scheme of *Taurinivorans muris* in the mouse gut.** *T. muris* mainly utilizes taurine as the main electron acceptor for anaerobic respiration in the gut but is also capable of thiosulfate and sulfolactate respiration. Pyruvate, lactate, and likely hydrogen are the main electron donors of *T. muris*, while formate could also be used. Taurine is cleaved from host-derived taurine-conjugated bile acids by other gut bacteria via bile salt hydrolase (BSH). Thiosulfate derives from mitochondrial oxidation of $H_2S$ in the gut epithelium. *T. muris* produces $H_2S$ from taurine via pyruvate-dependent taurine transaminase (Tpa), sulfoacetaldehyde (SA) acetyltransferase (Xsc), and dissimilatory sulfite reductase (DsrAB). $H_2S$ can have various effects on the gut microbiota and host health. For example, excess $H_2S$ can impair mucus integrity[3]. $H_2S$ can enhance resistance against enteropathogens by directly inhibiting enzymes in aerobically respiring *Klebsiella pneumoniae*[37]. *T. muris* could further impact microbial interactions and intestinal metabolism by stimulating the transcriptional activity of prophages that encode auxiliary metabolic genes, such as those involved in sulfur metabolism (S-AMG)[100]. Created with Biorender.com.

## LT0009-centric gut metatranscriptome analyses of laboratory mice

Cecal and fecal metatranscriptomes from a high-glucose diet experiment in mice (HG study) (JMF project JMF-2101-5) were analyzed for LT0009 gene expression. Mouse experiments were conducted following protocols approved by Austrian law (BMWF-66.006/ 0032-WF/ V/3b/2014). Additionally, mouse gut metatranscriptomes from a previous study were analyzed for LT0009 gene expression (Plin2 study)[101]. Sequence data (PRJNA379425) derived from eight-week-old C57BL/6 wild-type and Perilipin2-null (Plin2) mice fed high-fat/low-carbohydrate or low-fat/high-carbohydrate diets.

## Analyses of 16 S rRNA gene amplicon and metagenome data from Stacy et al. 2021

Metagenome and selected 16 S rRNA gene amplicon sequencing data of mouse models[91,102] that was shown in a previous study[91] to provide enhanced $H_2S$-mediated colonization resistance against the enteropathogens *Klebsiella pneumoniae* and/or *Citrobacter rodentium* were re-analyzed.

## Gnotobiotic oligo-mouse-microbiota mouse experiment

Animal experiments were approved by the local authorities (Regierung von Oberbayern; ROB-55.2-2532.Vet_02-20-84). Mice were housed in flexible film isolators (North Kent Plastic Cages) or Han-gnotocages (ZOONLAB) under germ-free conditions. The mice were supplied with autoclaved ddH₂O and Mouse-Breeding complete feed for mice (Ssniff) *ad libitum*. Twelve gnotobiotic C57BL/6 mice stably colonized with the 12-strain Oligo-Mouse-Microbiota (OMM[12]) community[72] were used for the animal experiment. OMM[12] mice ($n = 6$) were orally (50 μl)

and rectally (100 μl) inoculated with strain LT0009. The control OMM[12] mouse group ($n = 6$) was treated with the same volume of sterile 1x phosphate-buffered saline. After 10 days, the mice were infected with the human enteric pathogen *Salmonella enterica* serovar Typhimurium (avirulent *S. enterica* Tm strain M2702; $5×10^7$ c.f.u.) and sacrificed two days post infection (p.i.). Fecal microbiota composition was determined by strain-specific qPCR assays[72], including a newly developed assay for strain LT0009. Abundance of viable *S. enterica* Tm in feces and cecal content was determined by plating. Fecal samples from day two p.i. were used for (i) metatranscriptome sequencing ($n = 3$ for each group) at the Joint Microbiome Facility (JFM) of the Medical University of Vienna and the University of Vienna (JMF project JMF-2104-01) and (ii) taurine and bile acids quantification ($n = 3$ for each group).

## Mono- and co-colonization experiments in germ-free mice

Animal experiments were approved by the local authorities (Regierung von Oberbayern; ROB-55.2-2532.Vet_02-20-84). Seven female germ-free C57BL/6 mice were mono-associated with strain LT0009 ($n = 4$) or co-colonized with strain LT0009, *Bacteroides caecimuris* I48, and *Enterococcus faecalis* KB1 ($n = 3$) by gavage. Fecal samples were collected three and seven days after inoculation for strain-specific qPCR. All mice were sacrificed 10 days after initial inoculation. Intestinal content from ileum, cecum, colon, and feces was used for strain-specific qPCR and taurine and bile acids quantification.

## Bile acids and taurine quantification in mouse gut content

Metabolites were extracted from homogenized and dried gut content and fecal samples using methanol/acetonitrile/$H_2O$ (40:40:20; v:v:v) and reconstituted with (ACN/$H_2O$, 50:50, v-v). Targeted LC-MS/MS

analysis was performed on a Dionex Ultimate 3000 UHPLC (Thermo) system coupled to a TSQ Vantage triple quadrupole mass spectrometer (Thermo) via an electrospray ionization interface in negative polarity mode[103]. Each sample was measured in triplicate. The raw files obtained from the LC-MS/MS experiments were processed and quantified using the Skyline software[104]. The absolute concentrations of taurine and selected bile acids were calculated employing a dilution series of matrix samples spiked with reference standards of the analytes.

### Growth experiment of *E. clostridioformis*

*E. clostridioformis* YL32 was grown anaerobically in the brain-heart-infusion medium at 37 °C. Growth was monitored by spectrophotometric measurement of the optical density at 600 nm ($OD_{600}$). Sodium sulfide ($Na_2S$) was added to a final concentration of 0.5 mM or 5 mM. Sulfide was quantified as mentioned above. Host and prophage genes of strain YL32 were quantified by droplet digital PCR (ddPCR) (Supplementary Data 9).

### Statistics and reproducibility

All growth experiments were conducted with independent biological replicates and all replications revealed similar results. No statistical method was used to predetermine the sample size. No data were excluded from the analyses. The experiments were not randomized. The investigators were not blinded to allocation during experiments and outcome assessment.

### Reporting summary

Further information on research design is available in the Nature Portfolio Reporting Summary linked to this article.

## Data availability

Strain LT0009 has been deposited in the German Collection of Microorganisms and Cell Cultures (DSMZ) as DSM111569 and the Japan Collection of Microorganisms (JCM) as JCM34262. The genome and the 16 S rRNA gene sequence of *T. muris* LT0009 are available at NCBI GenBank under accession numbers CP065938 and MW258658, respectively. Sequencing data of the LT0009 pure culture transcriptome (JMF-2012-1) and the mouse gut metatranscriptomes from the HG study (JMF-2101-05) and the gnotobiotic study (JMF-2104-01) were deposited to the NCBI SRA under BioProject accession PRJNA867178. The mass spectrometry proteomics data have been deposited to the ProteomeXchange Consortium via the PRIDE[105] partner repository with the dataset identifier PXD044449. The LC-MS/MS data for taurine and bile acids quantification is publicly available at the Phaidra repository of the University of Vienna under the persistent identifier 1649944. All datasets used in this study are summarized in Supplementary Data 10. Source data are provided with this paper.

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

## Acknowledgements

We thank Bernhard Schink (University of Konstanz, Germany) for Latin naming, Daniela Gruber (Core Facility of Cell Imaging and Ultrastructure Research, University of Vienna) and Isabella Böhm for help with scanning electron microscopy, Markus Schmid for help with FISH imaging, and Jasmin Schwarz and Gudrun Kohl (Joint Microbiome Facility) for

sequencing. We also thank Holger Daims, Kerrin Steensen, Astrid Collingro, Hannes Schmidt, the DOME gut group members in Vienna as well as our colleagues at the University of Konstanz and LMU Munich for fruitful discussions and support. The mass spectrometric measurements were enabled by the Exposome/EIRENE Austria research infrastructure and the Mass Spectrometry Center of the Faculty of Chemistry at the University of Vienna. This work was financially supported by the Austrian Science Fund (FWF; project grants I2320-B22 and DOC 69-B to A.L.), the Deutsche Forschungsgemeinschaft (DFG; grants SCHL1936/3-4 to D.S., grants STE 1971/7-1, CRC1371, and P08 to B.S.), the Konstanz Research School Chemical Biology (KoRS-CB to B.S.), and the China Scholarship Council (Ph.D. fellowship grant no. 201606850092 to H.Y.).

## Author contributions

A.L. conceived the study. H.Y., B.T.H., P.P., B.S., and D.S. helped with the experimental design. B.S., B.W., and D.S. contributed essential experimental infrastructure and analytical instrumentation. H.Y., S.B., C.E., J.K., A.S.W., M.P., and B.T.H. performed experiments and analyzed data. B.Z. generated relevant preliminary data. H.Y., C.W.H., S.C., B.H., and P.P. performed bioinformatic analyses. S.B., A.S.W., J.K., B.T.H., P.P., M.P., B.W., D.S., and B.S. contributed to data interpretation. H.Y. and A.L. wrote the article. All authors discussed the results and revised the manuscript.

## Competing interests

The authors declare no competing interests.
