## [Peer Review File · Nature Communications]

REVIEWER COMMENTS

Reviewer #1 (Remarks to the Author):

Borusak and coworkers have obtained the first taurine-respiring murine bacterial isolate and have characterized it using a combination of in vitro, in vivo approaches and transcriptomic and proteomic analysis. In addition, they study the capacity of the isolate to confer colonization resistance against Salmonella. As I indicate in more detail below, the major conclusions obtained by this study are not well supported by the data and extensive additional experiments should be performed in order to support their conclusions. As it is right now, the manuscript does not provide a sufficient incremental advance for the broad readership of this journal.

- The authors investigate which genes are involved in taurine degradation by the novel identified bacterium. For this purpose, they analyze the transcriptome and proteome of the bacterial isolate grown using different electron acceptors, including taurine. They found that several genes, potentially related to taurine metabolism are expressed when taurine is administered. However, in most of the genes that the authors claim to be relevant for taurine degradation there is no difference in protein levels as compared to the growth of the bacteria using other electron acceptors. Thus the transcriptomic and proteomic data does not strongly suggest that these genes are involved in taurine metabolism. Moreover, none of the genes potentially acting in taurine degradation was experimentally validated in a more appropriate manner (i.e. obtaining mutants or expressing genes in other bacteria not capable of utilizing taurine). This is important because the putative function that was assigned in silico to a gene does not necessarily imply functionality in vivo. As an example, the authors indicate that the strain did not grow in cysteine despite expressing a gene (CuyA) that potentially metabolizes this compound.

- Using an in vivo model, the authors indicate that *T. muris* occupies in the intestine the vacant taurine niche of OMM12 mice. However this was only shown by transcriptomic data indicating that genes related to taurine metabolism are highly expressed. However this should be complemented with additional experimental data such as depletion or administration of taurine to mice and/or colonization with a mutant not capable of using taurine and investigate the impact in *T. muris* colonization. In addition the effect of *T. muris* administration on the levels of taurine and H₂S production in vivo should be evaluated.

- The authors claim that *T. muris* is protective against enteropathogens. However, the effect of *T. muris* against *Salmonella Typhimurium* in vivo is very limited. As the authors indicate, there is only a very slightly significant change in feces and this change is not significant in other parts of the large intestine (i.e. cecum). Also, what is the relevance of this change? Can the authors use an *S. typhimurium* strain that causes inflammation in mice and test if this slightly change in *Salmonella* levels is relevant? In addition, it will be important to understand the mechanisms of protection? Is it through inhibition of *Salmonella* respiration, through production of H₂S, as previously shown, or is it through nutrient competition? The authors again only show transcriptomic data to generate hypothesis but these hypotheses are not experimentally validated.

- The authors show that *T. muris* modifies the expression of genes encoded by a phage present in *E. clostridioformis* YL32 and discuss a potential mechanism of inducing expression (i.e. through H₂S production). However this is not validated experimentally. In addition, what is the relevance of this finding? Is there any phenotype associated to this increase in phage expression? Is this involved in colonization resistance against *Salmonella*?

- The authors indicate that LT0009 does not encode genes for bile salt hydrolase (BSH) and is thus likely dependent on other gut microbiota members for liberation of taurine. If so, is *T. muris* able to monocolonize GFM (in the absence of the OMM12 consortium)? Is *T. muris* colonization enhanced by OMM12? Does this depend on the capacity of the OMM12 bacteria to deconjugate bile acids?

Other points:

- Abstract "with protective capacity against pathogens". Only one pathogen was tested and the protection was minimal.

- Abstract "taurine respiration was the main in vivo lifestyle of *T. muris*". Not clear that this has been demonstrated - there is only expression and proteomic data and only with this data this conclusion is not strongly supported.

- Abstract: "*T. muris* selectively enhanced the activity of a sulfur metabolism gene-encoding prophage". This was only shown using transcriptomic/proteomic data. However, no experimental data about the activity of this gene has been shown.

- Abstract: "provided slightly increased colonization resistance against *Salmonella* Typhimurium, which showed reduced expression of galactonate catabolism genes". What is the impact of decreased expression of galactonate catabolism in the colonization levels of *Salmonella*?

- Abstract: "We identified *T. muris* as the dominant sulfidogen of a mouse microbiota?" Not sure about this since there is no experimental data regarding the levels of sulfide produced in the presence/absence of this bacteria or in different mouse colonies with different levels of this bacteria.

- Line 114: "The automated annotation of the genome was manually curated for genes of interest" What do the authors mean with manually curated? How was this performed?

- Line 115: "Phylogenomic analyses comprised treeing with 43 concatenated marker protein sequences" Which ones are these concatenated marker protein sequences? Can the authors supply this information as supplementary.

- Line 143: "Fecal microbiota composition was determined by strain-specific qPCR assays as previously described, including a newly developed assay for strain LT0009". Has this new assay been validated?

- Line 145: "Abundance of viable *S. enterica* Tm in feces and cecal content was determined by plating." Did the authors check that previous to the inoculation of *S. enterica* there was no growth of bacteria in the selective plates used for counting *S. enterica*?

- Line 174 "Strain LT0009 produced sulfide and acetate during taurine degradation" - Where are these results shown?
- Line 175 - "The genome comprises 2,059 protein ..." Please indicate this information, maybe as a supplementary table.
- Line 176 "free of contamination as assessed by CheckM" - can authors include a reference? how this was done? Is there any result from CheckM that can be shown to support the statement of "free of contamination"?
- Line 198: "SEM imaging further indicated that LT0009 cells have multiple polar flagella and are thus motile" Since authors have not demonstrate motility, please modify this sentence to "suggesting that they are motile"
- Line 204: "Both growth rate and final growth yields were increased when taurine was provided at 20 and 40 mmol/l concentration in comparison to 10 mmol/l" Are these the physiological concentrations found in the gut?
- Line 225: "differential transcriptomics and proteomics demonstrated that taurine is degraded via the Tpa-Xsc pathway and the produced sulfite respired via the DsrAB-DsrC pathway" But there is no higher levels of Xsc, DsrAB, DsrC at the proteomic level when using taurine?
- Line 313. "Notably, *T. muris*- and *B. wadsworthia*-related sequences co-occurred only in 28 mouse datasets, which suggests competitive exclusion possibly due to competition for taurine". This does not necessarily mean competitive exclusion for taurine. It could mean just that *T. muris* is more adapted to the mouse gut and that *B. wadsworthia* is more adapted to the human gut.
- Line 316. "we found that 82% of the *B. wadsworthia* positive samples are from mice that were 'humanized' by receiving human feces". These samples should not be included in the Figure 3 since this overestimates the real prevalence of *B. wadsworthia* in mice.

Reviewer #2 (Remarks to the Author):

Merits

This paper characterizes the bacterium *Taurinivorans muris*, the first novel taurine-respiring bacterium isolated from the mouse gut. Authors report that *T. muris* preferentially respire taurine via the Tpa-Xsc pathway, which diverges from the method of metabolism from the known taurine respirer *Bilophila wadsworthia*. Additionally, authors explore the distribution of this bacterium among different hosts and demonstrate that *T. muris* and *B. wadsworthia* reside in distinct ecological niches. Authors also show in a gnotobiotic mouse model that *T. muris* increased resistance from colonization from *S. enterica*. Overall, this is a well written paper of importance to researchers interested in microbial sulfur metabolism. Given the abundance of taurine as a metabolic substrate in the gut, it is important to define

bacteria capable of this metabolism particularly if their presence does not overlap. Below I have some minor critiques that could lend support to some of the hypotheses posited in the manuscript.

Critique

1. Major: What type of gas was used to grow the bacteria anaerobically? There is a paper that demonstrates that hydrogen is an important substrate for *Bilophila wasdworthia*. Just curious if this is also the case for this novel bacterium.
2. Major: Double check the references. Some of them are incomplete. (Example: Reference 76).
3. Minor: Please introduce all abbreviations in the manuscript.
4. Minor: Please state exactly how many datasets were explored for taurine-respiring bacteria.
5. Minor: There are paragraphs throughout that contain only one sentence. Suggest combining with other paragraphs.
6. Minor: I think the term "Taurine conjugated bile acids" is more correct than "taurocholic bile acids". In my mind, the latter refers to taurocholate specifically.
7. Major: "Our results suggest that the mouse commensal *T. muris* can enhance colonization resistance against 452 different enteropathogens." Did you test more than *S. enterica* Tm? Perhaps I am missing something.
8. Minor: A few missing words throughout text.
9. Minor: If you had cecal content left over, quantification of bile acids would be interesting in order to test your hypothesis regarding *T. muris* driving deconjugation.

Response to reviewers' comments

Reviewer #1 (Remarks to the Author):

Borusak and coworkers have obtained the first taurine-respiring murine bacterial isolate and have characterized it using a combination of in vitro, in vivo approaches and transcriptomic and proteomic analysis. In addition, they study the capacity of the isolate to confer colonization resistance against Salmonella. As I indicate in more detail below, the major conclusions obtained by this study are not well supported by the data and extensive additional experiments should be performed in order to support their conclusions. As it is right now, the manuscript does not provide a sufficient incremental advance for the broad readership of this journal.

>RESPONSE: We appreciate the critical comments of this reviewer, which we have used to improve our manuscript. We understand that some of our findings were not clearly and convincingly outlined. Therefore, we have clarified the key experimental findings below, performed further experiments and analyses that provide additional support for our discussion and conclusions, and revised the manuscript accordingly.

- The authors investigate which genes are involved in taurine degradation by the novel identified bacterium. For this purpose, they analyze the transcriptome and proteome of the bacterial isolate grown using different electron acceptors, including taurine. They found that several genes, potentially related to taurine metabolism are expressed when taurine is administered. However, in most of the genes that the authors claim to be relevant for taurine degradation there is no difference in protein levels as compared to the growth of the bacteria using other electron acceptors. Thus the transcriptomic and proteomic data does not strongly suggest that these genes are involved in taurine metabolism.

Moreover, none of the genes potentially acting in taurine degradation was experimentally validated in a more appropriate manner (i.e. obtaining mutants or expressing genes in other bacteria not capable of utilizing taurine). This is important because the putative function that was assigned in silico to a gene does not necessarily imply functionality in vivo. As an example, the authors indicate that the strain did not grow in cysteate despite expressing a gene (CuyA) that potentially metabolizes this compound.

>RESPONSE: We agree with the reviewer that gene knockouts, heterologous expression and/or biochemical protein characterization ultimately confirm the function of unknown genes/proteins. Yet, we are convinced that the most parsimonious explanation of our experimental results is that the Tpa, Ald, and Xsc homologs in *T. muris* are catalyzing taurine degradation.

First, the taurine-metabolism genes of *T. muris* are not novel, but show high sequence similarity and conserved functional amino acid residues to functionally confirmed Tpa, Ald, and Xsc enzymes (please see new subpanels for these enzymes in Supplementary Figure S4).

Second, our growth experiments prove that *T. muris* utilizes taurine, sulfolactate or thiosulfate for intracellular sulfite production, which is the ultimate electron acceptor for anaerobic respiration and energy conservation via the central DsrAB-DsrC sulfite reduction system. Fermentation was not observed.

Given that mRNA and protein pools can be incongruent^{2,3}, it was to be expected that differential mRNA and protein expression would not be significant for every single gene/protein in taurine, sulfolactate, and thiosulfate degradation. This is why we have analyzed both transcriptome and proteome of *T. muris*.

Please note that DsrAB-DsrC catalyzes the final, energy-conserving step in respiration of taurine, sulfolactate, and thiosulfate, and thus differential expression of the respective genes and proteins is not expected. Importantly, differential transcriptome and proteome analysis did not provide any hints for unknown proteins involved in taurine degradation.

The reviewer is right that Tpa, Ald, and Xsc are highly expressed, i.e., among the top 13 most abundant proteins, in *T. muris* grown with taurine, sulfolactate, and thiosulfate (see newly added Supplementary Figure S4). High prevalence of Tpa, Ald, and Xsc is likely due to their

constitutive expression and/or high protein stability. Yet, relative protein abundance of Tpa and Ald, but not Xsc, was significantly increased in cells grown with taurine compared to cells grown with thiosulfate. Furthermore, the expression of the key enzyme for sulfolactate desulfonation, SuyAB, was significantly increased in sulfolactate-grown cells compared to taurine- or thiosulfate-grown cells.

In conclusion, while Tpa, Ald, and Xsc are prevalent proteins independent of sulfur compound electron acceptor, growth on taurine and sulfolactate lead to a significant increase of Tpa/Ald and SuyAB, respectively, which clearly suggests a differential regulation of the two different organosulfonate metabolisms. Given that taurine is a permanently available substrate in the animal gut, a predominantly constitutive expression of taurine degrading proteins could sustain fitness of *T. muris* in the gut ecosystem⁴.

As mentioned in our manuscript, the function of the CuyA homolog in *T. muris* is unclear. Also because it can not only act as L-cysteate sulfo-lyase but also as D-cysteine desulfhydrase, which we now clarify in the revised manuscript (line 330).

In contrast to the constantly high mRNA expression levels (top 5% of all *T. muris* genes) of the taurine metabolism genes *tpa*, *ald*, and *xsc* *in vivo*, *cuyA* expression levels varied among different mouse gut samples, ranking from 8% to 46%. This suggests that CuyA is not directly involved in taurine metabolism. The actual physiological role of the CuyA homolog in *T. muris* will need to be investigated in future studies.

By including the additional results and information mentioned above and in the absence of permits from governmental authorities and available options to perform gene knockouts in *T. muris* or heterologous expression of *T. muris* proteins that would allow such an experiment to be performed in reasonable time, we have revised the wording in this manuscript section (lines 288-380).

- Using an *in vivo* model, the authors indicate that *T. muris* occupies in the intestine the vacant taurine niche of OMM12 mice. However this was only shown by transcriptomic data indicating that genes related to taurine metabolism are highly express. However this should be complemented with additional experimental data such as depletion or administration of taurine to mice and/or colonization with a mutant not capable of using taurine and investigate the impact in *T. muris* colonization. In addition the effect of *T. muris* administration on the levels of taurine and H₂S production *in vivo* should be evaluated.

>RESPONSE: We hope that the response to the reviewer's comments above, which partly also addresses these comments, provides some clarification. Additionally, we would like to re-stress that *tpa-xsc-dsrABC* were the most expressed genes in various mouse models and different dietary regimes. This high and permanent taurine metabolism gene transcription, in addition to the fundamental lack of alternative energy metabolisms (besides respiration of thiosulfate or sulfolactate, of which the latter is not known to occur in the gut), in our view, clearly shows the *in vivo* relevance of taurine respiration for *T. muris*.

While the development of a taurine-metabolism mutant of LT0009 and its *in vivo* application could be helpful in identifying a hypothetical, cryptic energy metabolism of *T. muris*, we feel that this is beyond the scope of the current study, given this is a new, fastidious strain for which genetic manipulation tools would need to be evaluated and/or developed. The narrow

substrate profile of this strain also poses significant challenges to taurine metabolism gene knockouts. Please also note that deletion knock-outs of *dsrABC* genes were lethal for all organisms tested so far^{5,6}.

Available samples from the gnotobiotic mouse experiment have not been preserved for H₂S quantification. Measurement of production and consumption rates of H₂S *in vivo*, which would be more informative than mere H₂S concentrations, is generally not straightforward. H₂S concentration in the gut is dependent on chemical and biological processes that produce and consume H₂S, which are manifold, and accurate measurements of H₂S are notoriously difficult⁷. Given the short-half life of mRNA in general and the correlation of cellular *dsrAB*-mRNA levels⁸ with cellular sulfate/sulfite reduction rates (=sulfide production rates), expression of *dsrAB* and *dsrC* is thus regarded as an appropriate proxy for active H₂S formation⁵. Please note that *dsrABC* genes are absent in the OMM¹² strains.

Furthermore, we have now analyzed the abundance of taurine and selected unconjugated and taurine-conjugated bile acids in available samples from the OMM¹² mouse experiments. LT0009-colonized mice showed a significant, 15-fold reduction in fecal taurine concentrations and a significant reduction of several taurine-conjugated bile acids (new Figure 4d). These new results support (i) our conclusion that *T. muris* LT0009 largely occupied the vacant taurine-nutrient niche in the intestinal tract of OMM¹² mice and (ii) our hypothesis that taurine degradation by LT0009 could provide a positive feedback mechanism on expression of bile salt hydrolases for taurine-conjugated bile acid deconjugation in the OMM¹² model.

As suggested by the reviewer (see below), we have also performed mono- and co-colonization experiments of *T. muris* in germ-free mice, which demonstrated that *T. muris* is strictly dependent on other, taurine-conjugated bile acids-deconjugating strains for colonization of the mouse gut (new Supplementary Figure S8).

We have revised the respective paragraphs in the manuscript (lines 458-466, 480-482)

- The authors claim that *T. muris* is protective against enteropathogens. However, the effect of *T. muris* against *Salmonella Typhimurium* *in vivo* is very limited. As the authors indicate, there is only a very slightly significant change in feces and this change is not significant in other parts of the large intestine (i.e. cecum). Also, what is the relevance of this change? Can the authors use an *S. typhimurium* strain that cause inflammation in mice and test if this slightly change in *Salmonella* levels is relevant? In addition, it will be important to understand the mechanisms of protection? Is it through inhibition of *Salmonella* respiration, through production of H₂S, as previously shown, or is it through nutrient competition? The authors again only show transcriptomic data to generate hypothesis but these hypothesis are not experimentally validated.

>RESPONSE: We were moderate in our interpretation of the magnitude of the effects of *T. muris* on colonization resistance against *S. enterica* Tm. Yet, we would like to stress that *S. enterica* CFUs were significantly reduced by about 50% in the presence of *T. muris* LT0009. This reduction is in the range of protective effects by addition of single- or multiple-strain consortia observed in other pathogen-mouse studies^{9,10}.

As mentioned in the manuscript, we investigated the impact of *T. muris* LT0009 during the initial niche invasion of *S. enterica* using an avirulent, non-colitogenic strain. Re-performing this experiment with a virulent *S. enterica* strain will be a relevant next step to address the additional impact of *T. muris* in latter phases of an infection with *S. enterica* and during inflammation, yet was beyond the scope of this study. We have revised the text to clarify that our metatranscriptome results do not provide evidence for H₂S-mediated inhibition of Salmonella respiration in early colonization. We had/have acknowledged in our manuscript that “The significance of galactonate for *S. enterica* Tm gut colonization and competition remains to be elucidated.” (Lines 515-516)

We have further shown that *T. muris* is the most abundant taurine-utilizing and H₂S producing member of the WildR mouse community that increased H₂S-mediated resistance to *K. pneumoniae* in the study by Stacy *et al.*¹. This is now further supported by an additional analysis of the metagenome data from this and a related study¹¹, showing that *T. muris* *dsrAB* sequences constitute, on average, 79% of all *dsrAB* genes in this community (new Figure 5).

(Lines 550-557)

- The authors show that *T. muris* modifies the expression of genes encoded by a phage present in *E. clostridioformis* YL32 and discuss a potential mechanism of inducing expression (i.e. through H₂S production). However this is not validated experimentally. In addition, what is the relevance of this finding? Is there any phenotype associated to this increase in phage expression? Is this involved in colonization resistance against Salmonella?

>RESPONSE: We have hypothesized that the increased H₂S production through presence of *T. muris* in the OMM¹² mouse model is a selective activator of the prophage Saumur in *E. clostridioformis* YL32. Increased transcriptional activity of this prophage was associated with increased colonization resistance against *Salmonella*.

We have now performed a first *in vitro* pure culture growth experiment with *E. clostridioformis* YL32 to test the hypothesis that sulfide is a prophage activator. These tests showed that addition of a high, physiological relevant dose of sulfide (5 mM Na₂S) reduced the growth of strain YL32, but did not activate its prophage Samur (see Supplementary information and new Supplementary Figure S10). The trigger of the transcriptional activation of this prophage in the gnotobiotic mice colonized by the OMM¹² strains and *T. muris* thus remains subject of future studies. We have revised the text of the manuscript accordingly (Lines 533-542).

- The authors indicate that LT0009 does not encode genes for bile salt hydrolase (BSH) and is thus likely dependent on other gut microbiota members for liberation of taurine. If so, is *T. muris* able to monocolonize GFM (in the absence of the OMM12 consortium)? Is *T. muris* colonization enhance by OMM12? Does this depend on the capacity of the OMM12 bacteria to deconjugate bile acids?

>RESPONSE: Thank you for this comment. We have now performed additional *in vitro* growth experiments with spent media and, as suggested, mono- and co-colonization experiments in germ-free mice. The results are summarized in a new section in the supplementary information, including the new supplementary figures S7 and S8. Briefly, we first confirmed that the LT0009 pure culture is not able to grow with taurocholic acid as substrate. However, spent medium produced by growing OMM¹² strains on medium with taurocholic acid contained free taurine and supported growth of LT0009 (new Supplementary Fig S7). In addition, LT0009

was not able to mono-colonize germ-free mice, but successfully co-colonized germ-free mice in combination with the taurine-conjugated bile acids-deconjugating strains *Bacteroides caecimuris* I48 and *Enterococcus faecalis* KB1 (new Supplementary Fig S8). We have revised the text of the manuscript accordingly (Lines 462-466).

Other points:

- Abstract "with protective capacity against pathogens". Only one pathogen was tested and the protection was minimal.

>RESPONSE: *T. muris* increased protection against *Salmonella enterica* and *Klebsiella pneumoniae*. Stacey *et al.*¹ previously showed that expansion of sulfidogenic commensals and the tpa-xsc-dsr pathway in the metagenome was fueled by host-derived taurine and increased protection against the enteropathogen *Klebsiella pneumoniae* and *Citrobacter rodentium* in mice; with sulfide-mediated inhibition of aerobic respiration by pathogens being proposed as a generic protective mechanism. As mentioned above and now included as a separate section in the main text, our re-analysis of the 16S rRNA gene amplicon data, and now also the metagenome data, from the study by Stacy *et al.*¹ identified *T. muris* as the dominant sulfidogen of the wildR community that protected against *K. pneumoniae* (new Fig. 5).

We have additionally revised the abstract and the conclusions for clarification (Line 573).

- Abstract "taurine respiration was the main in vivo lifestyle of *T. muris*". Not clear that this has been demonstrated - there is only expression and proteomic data and only with this data this conclusion is not strongly supported.

>RESPONSE: We hope that our response to the general comments above, additional experiments, and revision of the manuscript now provide sufficient support for this statement. We have also revised the wording in the abstract.

- Abstract: "*T. muris* selectively enhanced the activity of a sulfur metabolism gene-encoding prophage". This was only shown using transcriptomic/proteomic data. However, no experimental data about the activity of this gene has been shown.

>RESPONSE: Please note that this wording does not suggest activity of the gene, but the prophage harboring this gene. We have further modified the sentence in the abstract by referring to "transcriptional activity" for clarification.

- Abstract: "provided slightly increased colonization resistance against *Salmonella* Typhimurium, which showed reduced expression of galactonate catabolism genes". What is the impact of decrease expression of galactonate catabolism in the colonization levels of *Salmonella*?

>RESPONSE: As aforementioned, the potential impact of the reduced expression of galactonate catabolism genes in *Salmonella* remains subject of further studies.

- Abstract: "We identified *T. muris* as the dominant sulfidogen of a mouse microbiota?" Not sure about this since there is no experimental data regarding the levels of sulfide produced in the presence/absence of this bacteria or in different mouse colonies with different levels of this bacteria.

>RESPONSE: Please see our answers above regarding use of *dsrAB* mRNA levels as an adequate proxy for sulfidogenic activity.

- Line 114: "The automated annotation of the genome was manually curated for genes of interest" What do the authors mean with manually curated? How was this performed?

>RESPONSE: This information was/is available in the Supplementary Methods (SI Lines 177-186).

- Line 115: "Phylogenomic analyses comprised treeing with 43 concatenated marker protein sequences" Which ones are these concatenated marker protein sequences? Can the authors supply this information as supplementary.

>RESPONSE: Thank you for the comment. The reference tree used by CheckM was inferred from the concatenation of 43 conserved markers with largely congruent phylogenetic histories. The 43 markers are listed in supplementary table S6 from the CheckM publication of Parks et al.¹². The CheckM publication is now referenced in the methods sections of the main text and the supplement.

- Line 143: "Fecal microbiota composition was determined by strain-specific qPCR assays as previously described, including a newly developed assay for strain LT0009". Has this new assay been validated?

>RESPONSE: Yes. Thank you for the comment. We have now added the information on qPCR validation to the Supplementary Methods (SI Lines 341-355).

- Line 145: "Abundance of viable *S. enterica* Tm in feces and cecal content was determined by plating." Did the authors check that previous to the inoculation of *S. enterica* there was no growth of bacteria in the selective plates used for counting *S. enterica*?

>RESPONSE: Yes. The selective agar plates for *Salmonella enterica* showed no growth before plating and were kept sterile until usage.

- Line 174 "Strain LT0009 produced sulfide and acetate during taurine degradation" - Where are these results shown?

>RESPONSE: These results were/are shown in Figure 1d. Thank you for pointing out that a reference to the figure was missing. We have revised the manuscript (Lines 263-265).

- Line 175 - "The genome comprises 2,059 protein ..." Please indicate this information, maybe as a supplementary table.

>RESPONSE: The genome was annotated using the MicroScope annotation platform and we further used this annotation file for downstream analysis, such as transcriptome and proteome. All protein-encoding genes are listed in Supplementary Table 6 with the locus tags and annotation. For clarification, we now reference the supplementary table in this sentence: "The genome comprises 2,059 protein-coding genes (Supplementary Table 6), 56 tRNA genes, 4 rRNA operons (with 5S, 16S, and 23S rRNA genes), 4 pseudogenes, and 6 miscellaneous RNA genes." (Lines 236-238)

- Line 176 "free of contamination as assessed by CheckM" - can authors include a reference? how this was done? Is there any result from CheckM that can be shown to support the statement of "free of contamination"?

>RESPONSE: A reference for CheckM is now included (Parks, D. H., Imelfort, M., Skennerton, C. T., Hugenholtz, P. & Tyson, G. W. 2015. CheckM: assessing the quality of microbial genomes recovered from isolates, single cells, and metagenomes. *Genome Res.* 25, 1043– 1055). CheckM is a widely used standard tool to assess the quality of a genome assembly using a broader set of marker genes specific to the phylogenetic position of a genome within a reference genome tree and information about the collocation of these genes. CheckM results show percentages of contamination. For the strain LT0009 genome assembly, checkM showed contamination of 0%.

- Line 198: "SEM imaging further indicated that LT0009 cells have multiple polar flagella and are thus motile" Since authors have not demonstrate motility, please modify this sentence to "suggesting that they are motile"

>RESPONSE: Thank you. Done as suggested (Lines 260-261).

- Line 204: "Both growth rate and final growth yields were increased when taurine was provided at 20 and 40 mmol/l concentration in comparison to 10 mmol/l" Are these the physiological concentrations found in the gut?

>RESPONSE: Thank you for the question. Taurine concentrations for pure culture studies were based on previous physiological experiments with *B. wadsworthia* strains^{13–15}. Taurine concentrations in the gut are usually given as pmol or pg taurine per g of gut content/feces^{1,16} and thus these values can not be easily compared. However, by assuming that growth medium has the same density as water (1 liter = 1 kg, <https://www.biorxiv.org/content/10.1101/2020.08.25.266221v5>), 10-40 mmol/l taurine corresponds to 10-40 pmol/g. This is in the range of taurine concentrations we measured in the co-colonized germ-free mice (0.24 to 28.9 pmol/g) (new Supplementary Figure 8). Note that some of the free taurine was already utilized by *T. muris* LT0009 in these mice.

- Line 225: "differential transcriptomics and proteomics demonstrated that taurine is degraded via the Tpa-Xsc pathway and the produced sulfite respired via the DsrAB-DsrC pathway" But there is no higher levels of Xsc, DsrAB, DsrC at the proteomic level when using taurine?

>RESPONSE: Please see answer to general comments above. Briefly, DsrAB-DsrC are involved in respiration of taurine, sulfolactate, and thiosulfate, and thus differential expression of the respective genes and proteins is not expected. The reviewer is right that relative protein

abundance of Xsc was not significantly increased, but the two other taurine metabolism proteins Tpa and Ald were significantly increased in cells grown with taurine.

- Line 313. "Notably, *T. muris*- and *B. wadsworthia*-related sequences co-occurred only in 28 mouse datasets, which suggests competitive exclusion possibly due to competition for taurine". This does not necessarily mean competitive exclusion for taurine. It could mean just that *T. muris* is more adapted to the mouse gut and that *B. wadsworthia* is more adapted to the human gut.

>RESPONSE: Thank you. Differential adaptation to the human and mouse gut could also be a reason for this observation. We adapted the text accordingly (Lines 395-396).

- Line 316. "we found that 82% of the *B. wadsworthia* positive samples are from mice that were 'humanized' by receiving human feces". These samples should not be included in the Figure 3 since this overestimates the real prevalence of *B. wadsworthia* in mice.

>RESPONSE: We feel that a selective and complete removal of some datasets from this systematic analysis would bias the interpretation, as it also shows that *Bilophila* is able to successfully colonize mice. Instead, we have added an additional bar with the corrected data to the figure (Figure 3a) for visual illustration. We hope this clarifies the reviewers' concern.

Reviewer #2 (Remarks to the Author):

Merits

This paper characterizes the bacterium *Taurinivorans muris*, the first novel taurine-respiring bacterium isolated from the mouse gut. Authors report that *T. muris* preferentially respire taurine via the Tpa-Xsc pathway, which diverges from the method of metabolism from the known taurine respirer *Bilophila wadsworthia*. Additionally, authors explore the distribution of this bacterium among different hosts and demonstrate that *T. muris* and *B. wadsworthia* reside in distinct ecological niches. Authors also show in a gnotobiotic mouse model that *T. muris* increased resistance from colonization from *S. enterica*. Overall, this is a well written paper of importance to researchers interested in microbial sulfur metabolism. Given the abundance of taurine as a metabolic substrate in the gut, it is important to define bacteria capable of this metabolism particularly if their presence does not overlap.

Below I have some minor critiques that could lend support to some of the hypotheses posited in the manuscript.

>RESPONSE: We appreciate the reviewer's supportive words and comments, which we have used to improve our manuscript.

Critique

1. Major: What type of gas was used to grow the bacteria anaerobically? There is a paper that demonstrates that hydrogen is an important substrate for *Bilophila wadsworthia*. Just curious if this is also the case for this novel bacterium.

>RESPONSE: This information was/is provided in the supplement. Please see the section 'Additional energy metabolism of *Taurinivorans muris* LT0009' in the supplementary results & discussion for comparisons of *T. muris* and *B. wadsworthia* metabolisms of hydrogen and other possible electron donors for sulfite respiration (SI Lines 465-497). Briefly, *T. muris* LT0009 was grown in an anaerobic chamber under anoxic atmosphere with 85% N₂, 10% CO₂, and 5% H₂. We also specifically tested if using hydrogen as electron donor is able to support the growth of *T. muris* LT0009 (Figure 2b). This was not the case in our test condition (2 bar of hydrogen with additional acetate as carbon source).

2. Major: Double check the references. Some of them are incomplete. (Example: Reference 76).

>RESPONSE: Thank you. Corrected as suggested.

3. Minor: Please introduce all abbreviations in the manuscript.

>RESPONSE: Thank you. Done as suggested.

4. Minor: Please state exactly how many datasets were explored for taurine-respiring bacteria.

>RESPONSE: Done as suggested. The information of all datasets used in this study has now been included as new Supplementary Table 5.

5. Minor: There are paragraphs throughout that contain only one sentence. Suggest combining with other paragraphs.

>RESPONSE: We thank the reviewer for the suggestion. We merged most of the single sentence paragraphs with other paragraphs.

6. Minor: I think the term "Taurine conjugated bile acids" is more correct than "taurocholic bile acids". In my mind, the latter refers to taurocholate specifically.

>RESPONSE: We thank the reviewer for pointing this out. We changed "taurocholic bile acids" to "taurine-conjugated bile acids" throughout the manuscript.

7. Major: "Our results suggest that the mouse commensal *T. muris* can enhance colonization resistance against different enteropathogens." Did you test more than *S. enterica* Tm? Perhaps I am missing something.

>RESPONSE: Stacey *et al.*¹ previously showed that expansion of sulfidogenic Deltaproteobacteria commensals and the tpa-xsc-dsr pathway in the metagenome was fueled by host-derived taurine and increased protection against the enteropathogen *Klebsiella pneumoniae* in mice; with sulfide-mediated inhibition of aerobic respiration by pathogens being proposed as a generic protective mechanism. As discussed in the main text, our re-analysis of the 16S rRNA gene amplicon data from the wildR mouse model of the study by Stacey *et al.*¹ identified *T. muris* as the dominant deltaproteobacterium (Desulfobacterota) of the protective community.

We have now additionally re-analysed the metagenome data from this and a related study¹¹, which showed that *T. muris* *dsrAB* sequences constitute on average 79% of all *dsrAB* genes in this community (see the now extended Figure 5). This confirms that *T. muris* is the dominant taurine-reducing, H₂S-producing member of the wildR mouse community. This is now included as its own section in the main text (Lines 544-557). We have additionally revised the abstract and the conclusions for clarification (Line 573).

8. Minor: A few missing words throughout text.

>RESPONSE: We have checked the text, added some missing words, and hope our revision has addressed this comment.

9. Minor: If you had cecal content left over, quantification of bile acids would be interesting in order to test your hypothesis regarding *T. muris* driving deconjugation.

>RESPONSE: We thank the reviewer for the helpful suggestion. We have now analyzed the abundance of taurine and selected unconjugated and taurine-conjugated bile acids in available fecal samples from the OMM¹² mouse experiments. LT0009-colonized mice showed a significant, 15-fold reduction in fecal taurine concentrations and a significant reduction of several taurine-conjugated bile acids (new Figure 4d). These new results support (i) our conclusion that *T. muris* LT0009 largely occupied the vacant taurine-nutrient niche in the intestinal tract of OMM¹² mice and (ii) our hypothesis that taurine degradation by LT0009 could provide a selective feedback mechanism on expression of bile salt hydrolases for taurine-conjugated bile acid deconjugation in the OMM¹² model (Lines 458-462).

References

1. Stacy, A. *et al.* Infection trains the host for microbiota-enhanced resistance to pathogens. *Cell* **184**, 615–627.e17 (2021).
2. Smith, D. P. *et al.* Proteomic and transcriptomic analyses of ‘Candidatus Pelagibacter ubique’ describe the first PII-independent response to nitrogen limitation in a free-living Alphaproteobacterium. *MBio* **4**, e00133–12 (2013).
3. Heintz-Buschart, A. & Wilmes, P. Human gut microbiome: function matters. *Trends Microbiol.* **26**, 563–574 (2018).
4. Geisel, N. Constitutive versus responsive gene expression strategies for growth in changing environments. *PLoS One* **6**, e27033 (2011).
5. Santos, A. A. *et al.* A protein trisulfide couples dissimilatory sulfate reduction to energy conservation. *Science* **350**, 1541–1545 (2015).
6. Cort, J. R. *et al.* Allochromatium vinosum DsrC: solution-state NMR structure, redox properties, and interaction with DsrEFH, a protein essential for purple sulfur bacterial sulfur oxidation. *J. Mol. Biol.* **382**, 692–707 (2008).
7. Shen, X. *et al.* Measurement of plasma hydrogen sulfide in vivo and in vitro. *Free Radical Biology and Medicine* **50**, 1021–1031 (2011).

8. Neretin, L. N. *et al.* Quantification of dissimilatory (bi)sulphite reductase gene expression in *Desulfobacterium autotrophicum* using real-time RT-PCR. *Environ. Microbiol.* **5**, 660–671 (2003).
9. Pereira, F. C. *et al.* Rational design of a microbial consortium of mucosal sugar utilizers reduces *Clostridiodes difficile* colonization. *Nat. Commun.* **11**, 5104 (2020).
10. Jacobson, A. *et al.* A Gut commensal-produced metabolite mediates colonization resistance to *Salmonella* infection. *Cell Host & Microbe* **24**, 296–307.e7 (2018).
11. Rosshart, S. P. *et al.* Wild mouse gut microbiota promotes host fitness and improves disease resistance. *Cell* **171**, 1015–1028.e13 (2017).
12. Parks, D. H., Imelfort, M., Skennerton, C. T., Hugenholtz, P. & Tyson, G. W. CheckM: assessing the quality of microbial genomes recovered from isolates, single cells, and metagenomes. *Genome Res.* **25**, 1043–1055 (2015).
13. Laue, H. & Cook, A. M. Biochemical and molecular characterization of taurine:pyruvate aminotransferase from the anaerobe *Bilophila wadsworthia*. *Eur. J. Biochem.* **267**, 6841–6848 (2000).
14. Laue, H., Denger, K. & Cook, A. M. Taurine reduction in anaerobic respiration of *Bilophila wadsworthia* RZATAU. *Appl. Environ. Microbiol.* **63**, 2016–2021 (1997).
15. da Silva, S. M., Venceslau, S. S., Fernandes, C. L. V., Valente, F. M. A. & Pereira, I. A. C. Hydrogen as an energy source for the human pathogen *Bilophila wadsworthia*. *Antonie Van Leeuwenhoek* **93**, 381–390 (2008).
16. James, S. C. *et al.* Concentrations of fecal bile acids in participants with functional gut disorders and healthy controls. *Metabolites* **11**, (2021).

REVIEWER COMMENTS

Reviewer #1 (Remarks to the Author):

I really appreciate the efforts of the authors in replying to all my questions. They have performed new in vivo and in vitro experiments that provide novel results that strength their conclusions regarding taurine utilization by the identified bacteria and the relevance of other members of the microbiome in supplying taurine. I think this new results confirm that the identified bacteria is a major taurine utiliser in in the gut and have clearly improve this part of the manuscript. The results regarding the role of *T. muris* in colonization resistance against Enterobacteriaceae pathogens are not as strong as the previous ones. The authors provide novel data, mainly the re-analysis of a sequencing data available from a study that demonstrated a mechanism of colonization resistance against *K. pneumoniae* (i.e. sulfide production through taurine utilization) (Stacy et al.). In this analysis, the authors showed that in wild mice, the most abundant bacteria encoding genes for sulfide production is *T. muris*. However, other bacteria besides *T. muris* also encode for these genes (e.g. *Desulfovibrio*). Moreover, in the main mouse models used in the Stacy et al study to demonstrate the mechanisms of protection, the most abundant bacteria encoding for these genes are different (the family Peptococcaceae and *Desulfovibrio*). Thus, with this new, it cannot really be concluded that *T. muris* was the bacteria responsible for conferring protection against *K. pneumoniae*. It is possible, as shown in the Stacy et al. study, that the role of specific bacteria such as *T. muris* in conferring protection may depend on the intestinal environment (i.e. taurine levels which were increase upon infection and provide the nutrient source required for sulfide synthesis). If this is the case, and *T. muris* is the key bacteria conferring protection through this mechanism, administration of taurine to mice should boost *T. muris* resistance against *Salmonella*. Taking into account that one of the main messages of the manuscript is that *T. muris* contributes to colonization resistance against enteropathogens (the title of the manuscript), I think that the effect of *T. muris* in colonization resistance should be more convincing. One potential way of making this part of the study more robust could be to perform taurine supplementation experiments as the one suggested or testing the effect of *T. muris* against other enteropathogens, as I already indicated, including virulent ones that could demonstrate that the small effect observed in reducing the levels of *Salmonella* have any beneficial effect to the host.

Minor comments:

Suppl Fig. S7: Did the authors analyzed H₂S in SM and TSM media before LT0009 growth to verify that H₂S is not detected before LT0009?

The next sentence is a little bit confusing:

Expression of BSH genes in these strains did not change significantly with the presence of LT0009. Yet, BSH gene transcription increased in *Enterocloster clostridioformis* YL32, *Enterococcus* 450 faecalis KB1, *Bacteroides caecimuris* I48, and *Muribaculum intestinale* YL27, and decreased in 451 *Limosilactobacillus reuteri* I49.

Response to reviewers' comments

We thank the reviewer for providing positive feedback and constructive criticism on our manuscript. We have addressed all comments in our point-by-point response below and revised the manuscript accordingly.

Reviewer #1 (Remarks to the Author):

I really appreciate the efforts of the authors in replying to all my questions. They have performed new in vivo and in vitro experiments that provide novel results that strength their conclusions regarding taurine utilization by the identified bacteria and the relevance of other members of the microbiome in supplying taurine. I think this new results confirm that the identified bacteria is a major taurine utiliser in in the gut and have clearly improve this part of the manuscript.

>RESPONSE: We are happy and very much appreciate that the reviewer acknowledges our efforts and that the new results strengthen our conclusions on this part of the story.

The results regarding the role of *T. muris* in colonization resistance against Enterobacteriaceae pathogens are not as strong as the previous ones. The authors provide novel data, mainly the re-analysis of a sequencing data available from a study that demonstrated a mechanism of colonization resistance against *K. pneumoniae* (i.e. sulfide production through taurine utilization) (Stacy et al.). In this analysis, the authors showed that in wild mice, the most abundant bacteria encoding genes for sulfide production is *T. muris*. However, other bacteria besides *T. muris* also encode for these genes (e.g. *Desulfovibrio*). Moreover, in the main mouse models used in the Stacy et al study to demonstrate the mechanisms of protection, the most abundant bacteria encoding for these genes are different (the family Peptococcaceae and *Desulfovibrio*). Thus, with this new, it cannot really be concluded that *T. muris* was the bacteria responsible for conferring protection against *K. pneumoniae*.

>RESPONSE: This is not entirely correct and we would like to clarify this. Stacy et al. show sulfide-mediated protection against *Klebsiella pneumoniae* and/or *Citrobacter rodentium* in three mouse models: (1) the wildR model, (2) the $\Delta yopM$ model, and (3) in taurine supplemented mice. We show that *T. muris* is the most abundant *dsrAB*-encoding bacterium in wild mice AND wildR mice, which have received the microbiota from wild mice, while $\Delta yopM$ mice and taurine-supplemented mice have *Desulfovibrio* spp. and *Peptococcaceae* as the dominant *dsrAB*-encoding bacteria, respectively (Figure 5).

We have revised and extended the text to clarify that *T. muris* is the dominant sulfidogen in this particular protective mouse model but not in the others: "We identified *T. muris* as the dominant deltaproteobacterium (*Desulfobacterota*) and *dsrAB*-containing member of the microbiota of wild mice and the wild-mouse-microbiota (wildR) mouse model, but not in the other mouse models (Fig. 5). Given that taurine respiration via the sulfidogenic *tpa-xsc-dsr* pathway is the main energy niche of *T. muris* in the mouse gut (Fig. 4c, Supplementary Fig. 6), the enhanced resistance against *Klebsiella pneumoniae* in the wildR mouse model 37 was

thus primarily due to the activity of *T. muris*. Sulfide-mediated protection against *Klebsiella pneumoniae* and *Citrobacter rodentium* in the other mouse models, i.e., taurine-supplemented and $\Delta yopM$ mice, was not provided by *T. muris* but by other sulfidogens (Fig. 5) 37.”

It is possible, as shown in the Stacy et al. study, that the role of specific bacteria such as *T. muris* in conferring protection may depend on the intestinal environment (i.e. taurine levels which were increased upon infection and provide the nutrient source required for sulfide synthesis). If this is the case, and *T. muris* is the key bacteria conferring protection through this mechanism, administration of taurine to mice should boost *T. muris* resistance against *Salmonella*. Taking into account that one of the main messages of the manuscript is that *T. muris* contributes to colonization resistance against enteropathogens (the title of the manuscript), I think that the effect of *T. muris* in colonization resistance should be more convincing. One potential way of making this part of the study more robust could be to perform taurine supplementation experiments as the one suggested or testing the effect of *T. muris* against other enteropathogens, as I already indicated, including virulent ones that could demonstrate that the small effect observed in reducing the levels of *Salmonella* have any beneficial effect to the host.

>RESPONSE: We are convinced that the statements in our manuscript are correct and that our results show a protective role of *T. muris* against *Klebsiella* in the study by Stacy et al. and also point towards a protective role against *Salmonella*. Notably, the sulfide-mediated mechanism of protection against *Klebsiella* was essentially established by Stacy et al. Yet, we understand the concerns of the reviewer given that we do not clearly show the mechanism of protection against *Salmonella* in our gnotobiotic model. The suggested taurine supplementation experiment requires additional ethical approval by governmental authorities, and thus is not feasible for us to perform in a reasonable time. We have thus removed the emphasis on pathogen protection from the title and toned it down in the abstract. We hope that the reviewer finds this acceptable.

Minor comments:

Suppl Fig. S7: Did the authors analyze H₂S in SM and TSM media before LT0009 growth to verify that H₂S is not detected before LT0009?

>RESPONSE: Yes. Hydrogen sulfide in SM and TSM media was tested using a hydrogen sulfide test strip (Lead acetate test strips, Sigma-Aldrich, Cat. No. 70179). No hydrogen sulfide was detected in SM and TSM media before growth of LT0009.

The next sentence is a little bit confusing:

Expression of BSH genes in these strains did not change significantly with the presence of LT0009. Yet, BSH gene transcription increased in *Enterocloster clostridioformis* YL32, *Enterococcus faecalis* KB1, *Bacteroides caecimuris* I48, and *Muribaculum intestinale* YL27, and decreased in *Limosilactobacillus reuteri* I49.

>RESPONSE: The sentence has been rephrased and now reads as follows: “BSH gene transcription increased in *Enterocloster clostridioformis* YL32, *Enterococcus faecalis* KB1,

Bacteroides caecimuris I48, and *Muribaculum intestinale* YL27, and decreased in *Limosilactobacillus reuteri* I49, but not significantly (Supplementary Table 10).“